# EpiCoder: Encompassing Diversity and Complexity in Code Generation

**Yaoxiang Wang** [1 2 * †]  **Haoling Li** [3 * †]  **Xin Zhang** [4 *]  **Jie Wu** [3 †]  **Xiao Liu** [4]  **Wenxiang Hu** [4]  **Zhongxin Guo** [4]
**Yangyu Huang** [4]  **Ying Xin** [4]  **Yujiu Yang** [3]  **Jinsong Su** [1 2]  **Qi Chen** [4]  **Scarlett Li** [4]

## Abstract

Existing methods for code generation use code snippets as seed data, restricting the complexity and diversity of the synthesized data. In this paper, we introduce a novel feature tree-based synthesis framework, which revolves around hierarchical code features derived from high-level abstractions of code. The feature tree is constructed from raw data and refined iteratively to increase the quantity and diversity of the extracted features, which captures and recognizes more complex patterns and relationships within the code. By adjusting the depth and breadth of the sampled subtrees, our framework provides precise control over the complexity of the generated code, enabling functionalities that range from function-level operations to multi-file scenarios. We fine-tuned widely-used base models to obtain **EpiCoder** series, achieving state-of-the-art performance on multiple benchmarks at both the function and file levels. In particular, empirical evidence indicates that our approach shows significant potential in the synthesizing of repository-level code data. Our code and data are publicly available.[1]

## 1. Introduction

Large Language Models (LLMs) (OpenAI, 2023; Zhang et al., 2022) have demonstrated significant potential in the field of code understanding and generation, particularly

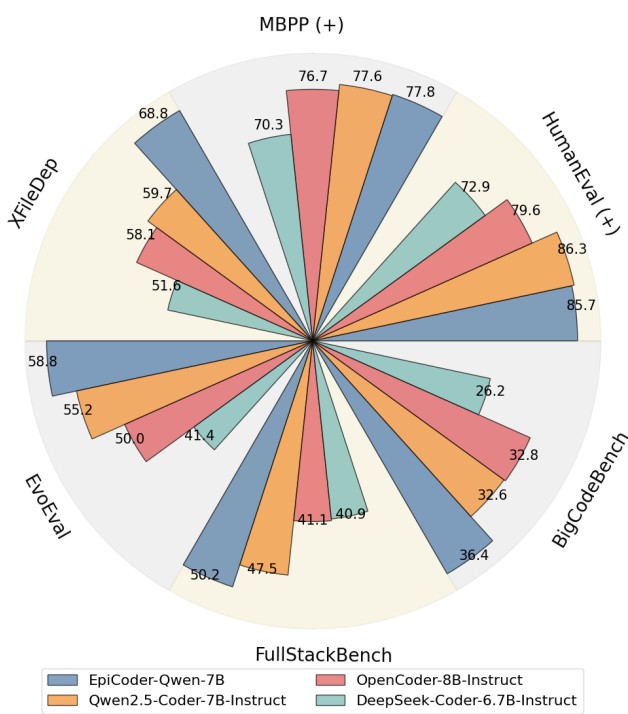

Figure 1: Benchmark performance of EpiCoder-Qwen-7B (fine-tuned on Qwen2.5-Coder-7B-Base) and its counterparts. XFileDep is file-level code generation benchmark, all others are function-level.

through pre-training on large-scale code data (Sun et al., 2024; Zhu et al., 2024). However, the latent code knowledge in these models is often underutilized without fine-tuning on high-quality instruction data. Instruction tuning has emerged as a critical step in unlocking the full capabilities of code LLMs (Wei et al., 2022), enabling a more precise alignment with user intent and improving the usability of the model in real-world scenarios.

Despite recent advances, most existing methods for synthesizing high-quality instruction data rely on code snippets as seed data (Chaudhary, 2023; Yu et al., 2023b; Wei et al., 2024b; Muennighoff et al., 2023). While code snippets demonstrate specific functionalities, they fail to capture the full range of programming constructs, patterns, and interactions that are common in real-world programming scenarios.

---

[*]Equal contribution [†]This work is done during their internships at Microsoft [1]School of Informatics, Xiamen University [2]Key Laboratory of Digital Protection and Intelligent Processing of Intangible Cultural Heritage of Fujian and Taiwan (Xiamen University), Ministry of Culture and Tourism, China [3]Tsinghua University [4]Microsoft. Correspondence to: Xin Zhang <xinzhang3@microsoft.com>, Yujiu Yang <yang.yujiu@sz.tsinghua.edu.cn>, Jinsong Su <jssu@xmu.edu.cn>.

*Proceedings of the $42^{nd}$ International Conference on Machine Learning*, Vancouver, Canada. PMLR 267, 2025. Copyright 2025 by the author(s).

[1]https://github.com/microsoft/EpiCoder and https://github.com/DeepLearnXMU/EpiCoder

Additionally, due to the inherent rigidity of code snippets, it is difficult to rearrange or recombine them to generate new and diverse combinations. These limitations restrict the overall complexity and diversity of the synthesized data, highlighting the critical need for more structured representations as seed data to overcome these constraints and address real-world programming challenges.

Inspired by Abstract Syntax Tree (AST), we propose a **feature tree-based** code data synthesis framework that revolves around hierarchical code features derived from high-level abstractions such as variable types function structures, and control flow. While AST captures the syntactic structure of code, our approach extends this idea by organizing features into a tree structure that captures the semantic relationships between code elements.

Specifically, we extract features from the seed data and use hierarchical clustering to generate a tree structure demonstration, starting from a feature set and proceeding bottom-up. This demonstration serves as a guideline for the LLM to directly extract tree structures from raw code data. To ensure comprehensive coverage of real-world scenarios, we enhance the diversity of features by iteratively expanding the tree structure both in breadth and in depth. Compared to methods that evolve based on individual code snippets or single instructions (Luo et al., 2023), our approach is more efficient and achieves broader coverage, as the tree structure provides clear and organized directions for evolution. The resulting feature tree is a large and hierarchical structure, from which we can sample subtrees to generate code data. Our feature tree-based synthesis method offers two advantages: (1) **Controllable Complexity**: Our framework allows for adjusting the complexity of synthesized data by modifying the depth and breadth of subtrees. This enables the creation of code ranging from simple function-level tasks to comprehensive, multi-file solutions. (2) **Targeted Learning**: By adjusting the probability of sampling features, we can prioritize specific knowledge areas that are underrepresented, ensuring a more tailored and effective learning process for the LLMs.

We conduct extensive experiments to validate our feature tree-based code data synthesis framework, training Qwen2.5-Coder-7B-Base and DeepSeek-Coder-6.7B-Base to derive the EpiCoder series model. Our EpiCoder-Qwen-7B trained on 380k function-level and 53k file-level data, respectively. Regarding the five function-level code generation benchmarks, it has achieved state-of-the-art (SOTA) performance on average among models of similar scales. Notably, in the completion tasks of BigCodeBench-Hard, it outperforms Qwen2.5-Coder-7B-Instruct by a margin of 7.4%. Furthermore, on file-level benchmark XFileDep, EpiCoder exhibits superior performance, underscoring its capability to address programming problems of varying com-

plexity. Building on these results, our approach further demonstrates strong potential in synthesizing highly complex repository-level data, as illustrated in Figure 5. To assess data complexity, we incorporated definitions from software engineering principles and employed the LLM-as-a-judge methodology. Moreover, we examined the diversity of datasets from a feature-based perspective to underscore the versatility and robustness of our framework.

Our contributions are summarized as follows:

- We propose a feature tree-based code synthesis framework that enables controllable synthesis of diverse and complex instruction data, ranging from function-level to file-level tasks.

- We synthesize 433k instruction data and train EpiCoder. Our EpiCoder-Qwen-7B achieves state-of-the-art performance among comparably sized models in multiple function-level and file-level benchmarks, demonstrating its capability to tackle programming problems of varying complexity.

- By conducting further analysis, we showcase the advantages of our data in terms of complexity and diversity, as well as its potential for synthesizing large-scale repositories.

## 2. Methodology

In this section, we present our feature tree-based code generation framework, which consists of three key steps: (1) **Feature Tree Extraction** (Section 2.1), where we construct the tree demonstration and extract feature trees from seed data; (2) **Feature Tree Evolution** (Section 2.2), where we iteratively expand the feature tree to enhance diversity and coverage; and (3) **Feature Tree-Based Code Generation** (Section 2.3), where we use the evolved feature tree to generate diverse code instruction data with varying complexity. An overview of the framework is illustrated in Figure 2.

### 2.1. Feature Tree Extraction

Inspired by Abstract Syntax Trees (AST), which represent the syntactic relationships in code, we use the LLM to extract a hierarchical representation that organizes key elements of code into a tree structure to capture more fundamental semantic relationships.

**Raw Code Collection** To ensure data diversity and comprehensiveness, we obtain seed data from The Stack v2 (Lozhkov et al., 2024), a publicly available large-scale dataset widely used for pre-training code LLMs. Following (Yu et al., 2023b), we apply the KCenterGreedy algorithm (Sener & Savarese, 2018) to select a core set of

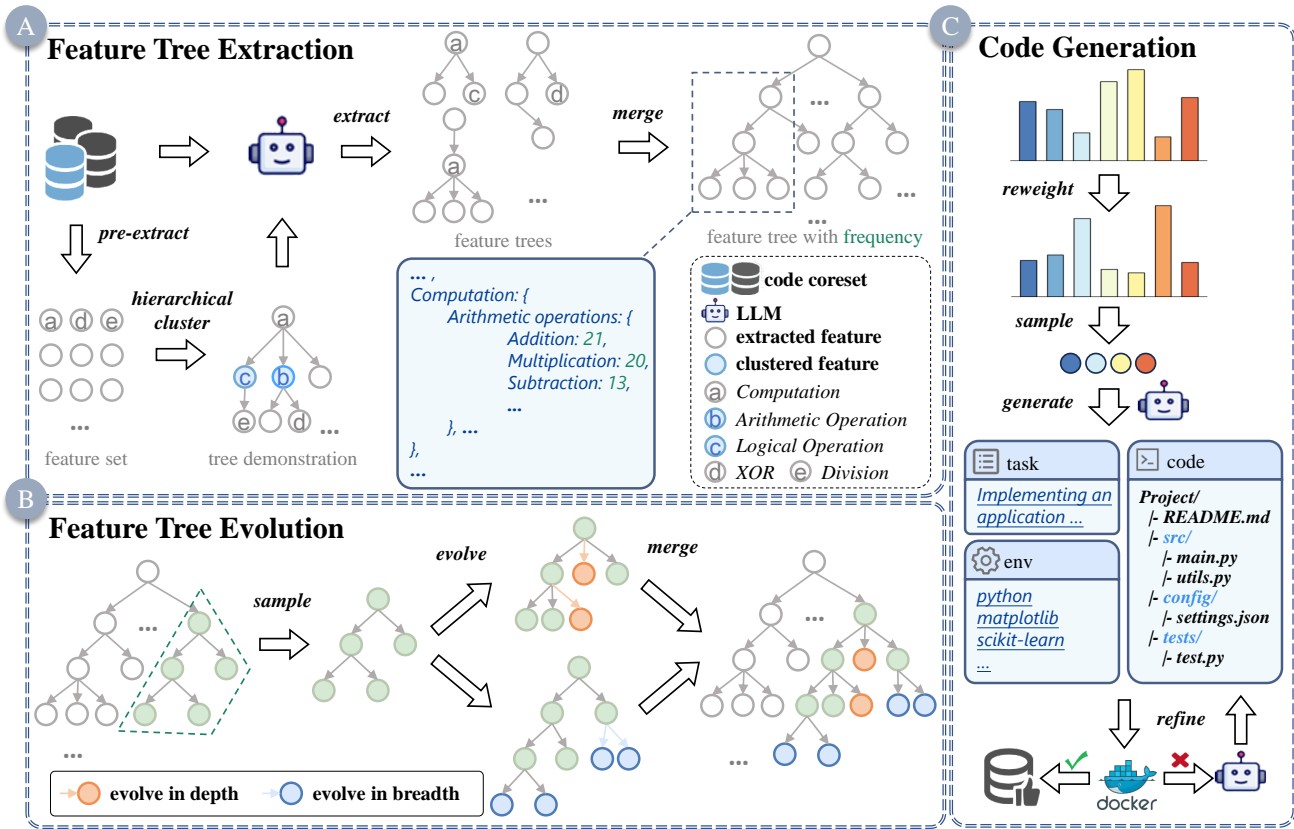

Figure 2: Overview of our feature tree-based code generation framework, which consists of three steps: (a) **Feature Tree Extraction**, where we first extract the feature set to construct the tree structure demonstration and then extract the feature trees; (b) **Feature Tree Evolution**, where the feature tree is iteratively expanded in depth and breadth; and (c) **Feature Tree-Based Code Generation**, where the evolved feature tree is used to generate diverse code instruction data. A detailed example of feature evolution and code generation is shown in Appendix A.

diverse samples based on the code embeddings encoded by roberta-large-v1 (Liu et al., 2019).

**Tree Demonstration Construction**    To extract features from the seed data, we leverage a powerful large language model (LLM), specifically GPT-4o. [2] Since the initial prompt provided to the LLM can significantly impact its response, we therefore propose an iterative method to optimize the demonstration tree structure within the prompt. As shown in Figure 2 (a), the construction of the demonstration consists of the following two steps:

1. **Feature Pre-Extraction:** With a draft version of the prompt, the LLM is used to extract an initial set of features from the seed data, producing a collection of feature keywords.

2. **Iterative Demonstration Generation:** For each step, we take a subset of the feature set and prompt the

LLM to perform hierarchical clustering, producing a tree structure that represents the relationships among the features. We iteratively refine the tree structure demonstration by adjusting the clustering of features, ensuring a well-organized hierarchical relationship.

**Feature Tree Extraction**    Using the refined demonstration of the tree structure, the LLM is tasked with extracting a tree-structured feature representation for each code snippet. These individual feature trees are then merged into a comprehensive structure that unifies the extracted features across all code snippets. During this process, we record the frequency of each node to reflect the distribution of features in the seed data. Since the seed data is derived from the pre-training data, this frequency can serve as an approximate measure of the knowledge distribution within the pre-trained model.

### 2.2. Feature Tree Evolution

To overcome the limitations in both diversity and quantity of features in the seed data, we expand the feature tree through

---

[2]Unless otherwise specified, the strong LLM refers to GPT-4o.

an evolutionary process. Compared to evolving directly from the seed code or instructions, this approach ensures broader feature coverage and improves efficiency by leveraging the tree's structured representation, which provides clear and systematic directions for evolution. As illustrated in Figure 2 (b), at each iteration, a subtree is sampled from the full feature tree. This subtree is then evolved by the LLM along two dimensions, depth and breadth, by adding finer-grained child nodes to existing nodes as well as sibling nodes at the same hierarchical level. These newly evolved subtrees are then merged back into the overall structure, significantly enriching the feature space and facilitating the generation of more diverse and complex code. An example of the evolution of a single subtree is provided in Appendix A.3.

One key challenge in feature evolution is to estimate the frequency of newly generated features. Unlike the feature extraction stage, where frequencies are calculated directly, the frequency of an evolved feature is estimated as the average frequency of its siblings. This approach reasonably estimates the frequency of new features according to the existing feature distribution, ensuring that evolved features integrate seamlessly into the broader feature tree.

### 2.3. Feature Tree-Based Code Generation

Traditional code generation methods that rely solely on code snippets often produce outputs that closely resemble the original seed data, limiting diversity and complexity. We mitigate this issue by generating code based on feature trees and the process is outlined as follows.

**Distribution Reweighting** The previously recorded feature frequencies partially reflect the distribution of natural data, helping to simulate real-world scenarios. However, some high-frequency but easy features, such as `config` and `initialize`, do not require strong focus during instruction tuning. To address this issue, we adjust the probability distribution of a node's child features:

$$p_i' = \frac{\exp(\log p_i/t)}{\sum_{j \in C} \exp(\log p_j/t)} \tag{1}$$

where $p_i$ represents the normalized original frequency of the child feature $i$ and $p_i'$ is the adjusted probability. As detailed in Algorithm 1, the summation applies to all child features $j$ in the set $C$, which denotes the set of child nodes for the current parent node. A higher temperature value $t$ leads to a smoother distribution, allowing less dominant features a higher probability of being selected. To further enhance the diversity of the generated data, we employed multiple temperature values during the data synthesis process for a wider range of feature distributions.

**Feature Sampling** To generate diverse code instruction data, we sample a subtree of candidate features from the

---

**Algorithm 1** Feature Sampling for Task Generation

1: **Input:** Current root node $R$, frequency library $F$, temperature $t$, sample size $S$
2: **Output:** Selected feature set
3: selected_set $\leftarrow \emptyset$
4: **for** $s = 1$ to $S$ **do**
5: $\quad C \leftarrow$ get_children$(R)$
6: $\quad$ **if** $C = \emptyset$ **then**
7: $\quad\quad$ **break**
8: $\quad$ **end if**
9: $\quad f_i \leftarrow F[i]$ for all $i \in C$
10: $\quad p_i \leftarrow \frac{f_i}{\sum_{j \in C} f_j}$ for all $i \in C$
11: $\quad$ Compute $p_i'$ using Equation (1) for all $i \in C$
12: $\quad$ cur_node $\leftarrow$ sample_node$(C, [p_1', p_2', \ldots])$
13: $\quad$ selected_set.add(cur_node)
14: **end for**
15: **Return** selected_set

---

feature tree according to the adjusted probability distribution. The sampling process is guided by a predefined shape of the subtree to sample, where we recursively apply Algorithm 1 to get the subtree. The LLM then uses this sampled subtree to generate code instruction data. By adjusting the depth and breadth of the subtree to sample, we can flexibly create tasks with varying complexity, thereby providing a broad spectrum of coding challenges.

**Content Generation** Based on the sampled subtree, the LLM proceeds to select a compatible subset and then generate a task and the corresponding code and execution environment. The solution code can range from a single function to a comprehensive multi-file project, depending on the task's complexity. By supporting multi-file solutions, this approach enables the generation of code that reflects a realistic project structure and effectively captures cross-file information for problems requiring interactions across different components. As illustrated in Figure 6 of Appendix A.1, different files implement distinct functionalities and collectively form an integrated system through their interdependencies.

**Iterative Refinement** To improve the quality of the generated data, the solution code is accompanied by relevant test files, which are also generated by the LLM. These tests are executed in an isolated environment, allowing us to identify and filter out incorrect solutions. Through an iterative debugging process, information from the execution results is used to guide the LLM to refine the solution, ensuring the correctness of the generated code.

# 3. Experiment

In this section, we introduce the details of the synthetic data (3.1) and evaluate the model's performance in code generation at different levels. Specifically, in Section 3.2, we evaluate the model's ability using five function-level code generation benchmarks. In Section 3.3, we employ our meticulously crafted XFileDep benchmark to evaluate the model's file-level code generation capabilities. We demonstrate the potential ability to generate particularly complex code repositories in Section 3.4.

## 3.1. Implementation Details

We utilized our pipeline to extract 5k features from the core set of 150k Python language files in The Stack V2. These features were then expanded through evolutionary methods to 140k features. Then we synthesized 380k function-level data samples and 53k file-level data samples based on the features. We choose DeepSeek-Coder-Base-6.7B (Guo et al., 2024) and Qwen2.5-Coder-7B (Hui et al., 2024) as the base LLMs and obtain the EpiCoder-DS-6.7B and EpiCoder-Qwen-7B models after training. To ensure a fair comparison with other baselines, we incorporated the evol-codealpaca-v1[3] (applied the same filtering criteria as described in (Yu et al., 2023b)) dataset into the training of only the DeepSeek-Coder-Base-6.7B model. Additionally, for models trained solely on file-level data, we employed additional notation. We evaluated the models on benchmarks corresponding to their respective training levels.

## 3.2. Function-level Generation

Many previous code LLMs have exhibited overfitting to specific benchmarks after fine-tuning, which somewhat constrains their ability to generalize to other benchmarks. To prevent this, we employed five benchmarks for evaluation, with HumanEval (Chen et al., 2021), MBPP (Austin et al., 2021), BigCodeBench (Zhuo et al., 2025), EvoEval (Xia et al., 2024) and FullStackBench (Liu et al., 2024), ensuring that they are generally broad, comprehensive, reliable, and decontaminated. For further clarity, all benchmarks are described in detail in Appendix B.1.

Table 1 and Figure 3 illustrate the performance of our model in these programming tasks at the function level. Except for the results underlined, which are sourced from their respective papers, all other results are obtained from the EvalPlus-Leaderboard[4] and BigCodeBench-Leaderboard[5]. Among models of the

---

[3]https://huggingface.co/datasets/theblackcat102/evol-codealpaca-v1
[4]https://evalplus.github.io/leaderboard.html
[5]https://huggingface.co/spaces/bigcode/bigcodebench-leaderboard

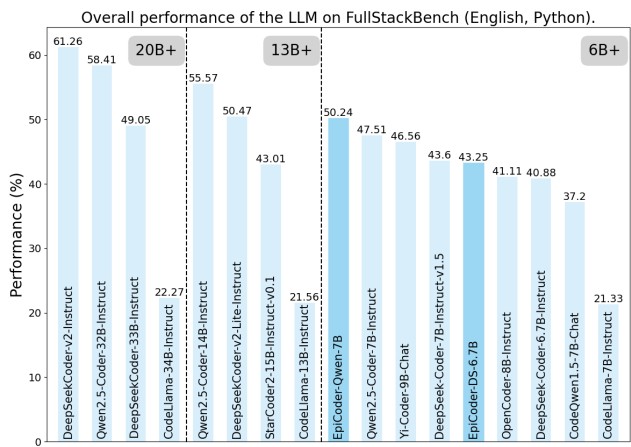

Figure 3: Model performance across domains of Python in the English Subset of FullStackBench.

same size, EpiCoder-Qwen-7B achieves the state-of-the-art (SOTA) average performance. The evaluation of the EpiCoder series models on these benchmarks highlights their capability to solve challenge and complex programming problems. Moreover, the results also demonstrate that the feature tree-based code generation method can provide high-quality and diverse synthetic data tailored to function-level programming problems.

## 3.3. File-level Generation

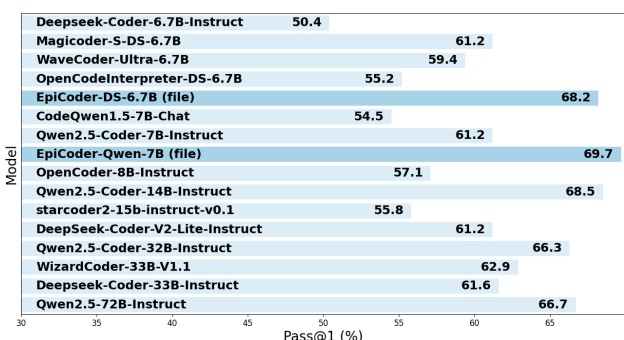

Figure 4: Pass@1 (%) results of different LLMs on XFileDep computed with greedy decoding.

**XFileDep Benchmark** Many existing code benchmarks focus on function-level code generation and lack evaluations of file-level generation capabilities. To address this limitation, we have developed a Cross-File Dependency Benchmark (XFileDep) specifically designed to assess the file-level code generation capabilities of LLMs while considering cross-file dependencies. XFileDep provides a comprehensive framework by treating multiple interdependent code files as context and testing the model's ability to generate missing files. This benchmark not only measures the model's ability to generate isolated files, but also evaluates

Table 1: Pass@1 (%) results of different LLMs on HumanEval (+), MBPP (+) and BigCodeBench computed with greedy decoding. We conducted the evaluation on the **Full** and **Hard** subsets of BigCodeBench (BCB), including the *Complete (Comp.)* and *Instruct (Ins.)* tasks.

| Model | Base | HumanEval | | MBPP | | BCB-Full | | BCB-Hard | | EvoEval | Average |
| | | *Base* | *Plus* | *Base* | *Plus* | *Comp.* | *Ins.* | *Comp.* | *Ins.* | | |
| Closed-source Model | | | | | | | | | | | |
| GPT-4-Turbo (April 2024) | - | 90.2 | 86.6 | 85.7 | 73.3 | 58.2 | 48.2 | 35.1 | 29.1 | 61.4 | 63.1 |
| Claude-3.5-Sonnet (June 2024) | - | 87.2 | 81.7 | 89.4 | 74.3 | 58.6 | 46.8 | 33.1 | 25.7 | - | - |
| 7B+ Scale | | | | | | | | | | | |
| Qwen2.5-Coder-32B-Instruct | - | 92.1 | 87.2 | 90.5 | 77.0 | 58.0 | 49.0 | 33.8 | 27.7 | 67.6 | 64.8 |
| OpenCoder-8B-Instruct | - | 81.7 | 77.4 | 82.0 | 71.4 | 50.9 | 43.2 | 18.9 | 18.2 | 49.2 | 54.8 |
| DeepSeek-Coder-33B-instruct | - | 81.1 | 75.0 | 80.4 | 70.1 | 51.1 | 42.0 | 20.9 | 17.6 | 51.6 | 54.4 |
| Codestral-22B-v0.1 | - | 79.9 | 73.8 | 72.5 | 61.9 | 52.5 | 41.8 | 24.3 | 16.9 | 57.2 | 53.4 |
| ∼ 7B Scale | | | | | | | | | | | |
| 🐳 DSCoder-6.7B-Base | - | 47.6 | 39.6 | 72.0 | 58.7 | 41.8 | - | 13.5 | - | 30.4 | - |
| DeepSeekCoder-6.7b-Instruct | 🐳 | 74.4 | 71.3 | 74.9 | 65.6 | 43.8 | 35.5 | 15.5 | 10.1 | 41.4 | 48.1 |
| Magicoder-S-DS | 🐳 | 76.8 | 71.3 | 79.4 | **69.0** | 47.6 | 36.2 | 12.8 | **13.5** | 44.6 | 50.1 |
| WaveCoder-Ultra-6.7B | 🐳 | 75.0 | 69.5 | 74.9 | 63.5 | 43.7 | 33.9 | 16.9 | 12.8 | 42.0 | 48.0 |
| OpenCodeInterpreter-DS-6.7B | 🐳 | 77.4 | 72.0 | 76.5 | 66.4 | 44.6 | 37.1 | 16.9 | **13.5** | 44.2 | 49.8 |
| **EpiCoder-DS-6.7B** | 🐳 | **80.5** | **76.8** | **81.5** | 68.3 | **50.6** | **37.9** | **19.6** | 12.8 | **50.0** | **53.1** |
| 🐦 Qwen2.5-Coder-7B-Base | - | 61.6 | 53.0 | 76.9 | 62.9 | 45.8 | - | 16.2 | - | 35.6 | - |
| Qwen2.5-Coder-7B-Instruct | 🐦 | 88.4 | **84.1** | 83.5 | **71.7** | 48.8 | 40.4 | 20.3 | 20.9 | 55.2 | 57.0 |
| **EpiCoder-Qwen-7B** | 🐦 | **89.0** | 82.3 | **84.1** | 71.4 | **51.9** | **43.8** | **27.7** | **22.3** | **58.8** | **59.0** |

its proficiency in understanding and managing cross-file dependencies. The detailed process for constructing the benchmark is described in Appendix B.2.

**Evaluation Details** The XFIleDep benchmark comprises a total of 466 problems. For all problems, we employed pass@1 as the evaluation metric, as we believe that only code passing all test cases can be considered to be functioning correctly. To ensure stable evaluation, we used docker to standardize the running environment for all problems. During the testing of different models, we consistently applied their default prompts and greedy decoding, with a maximum token length of 8192. All open-source models were accelerated using vLLM (Kwon et al., 2023). Figure 4 shows the results of different LLMs on this benchmark. The experimental results indicate that the EpiCoder, which have been specifically trained on a dataset consisting of over 53k multi-file code samples, significantly outperform the baseline models. This demonstrates the EpiCoder's exceptional capability in generating file-level code while simultaneously considering cross-file dependencies. Furthermore, it validates our approach's advantage in synthesizing multi-file code data of higher complexity.

### 3.4. Potential Repo-level Generation

Our approach benefits from a hierarchical structure of the feature tree, enabling the synthesis of instructions and corresponding outputs of varying complexities. We further

explore its limit by attempting to synthesize much more complex real-world code repository data than the file-level data that contains only several files. Specifically, utilizing a feature tree extracted from the popular open-source GitHub repository LLaMA-Factory[6], we successfully generate a repository mirroring the structure with over 50 files. The example files within this repository demonstrate the feasibility and scalability of our approach, as illustrated in Figure 5. This example highlights the potential of feature-based code generation to produce complex and structured repositories, offering a promising direction for future research in repository-level code synthesis.

## 4. Further Analysis

In this section, we first analyze the complexity of the generated code (Section 4.1), and then we evaluate the diversity of the code based on feature analysis (Section 4.2). Additionally, we investigate the scaling effect of code instruction data and assess potential data leakage issues, which are detailed in Appendix C.

### 4.1. Complexity Evaluation

We emphasize that our ability to generate more complex code data stems from its hierarchical feature tree structure. By flexibly adjusting the depth and width of the feature tree,

---

[6] https://github.com/hiyouga/LLaMA-Factory

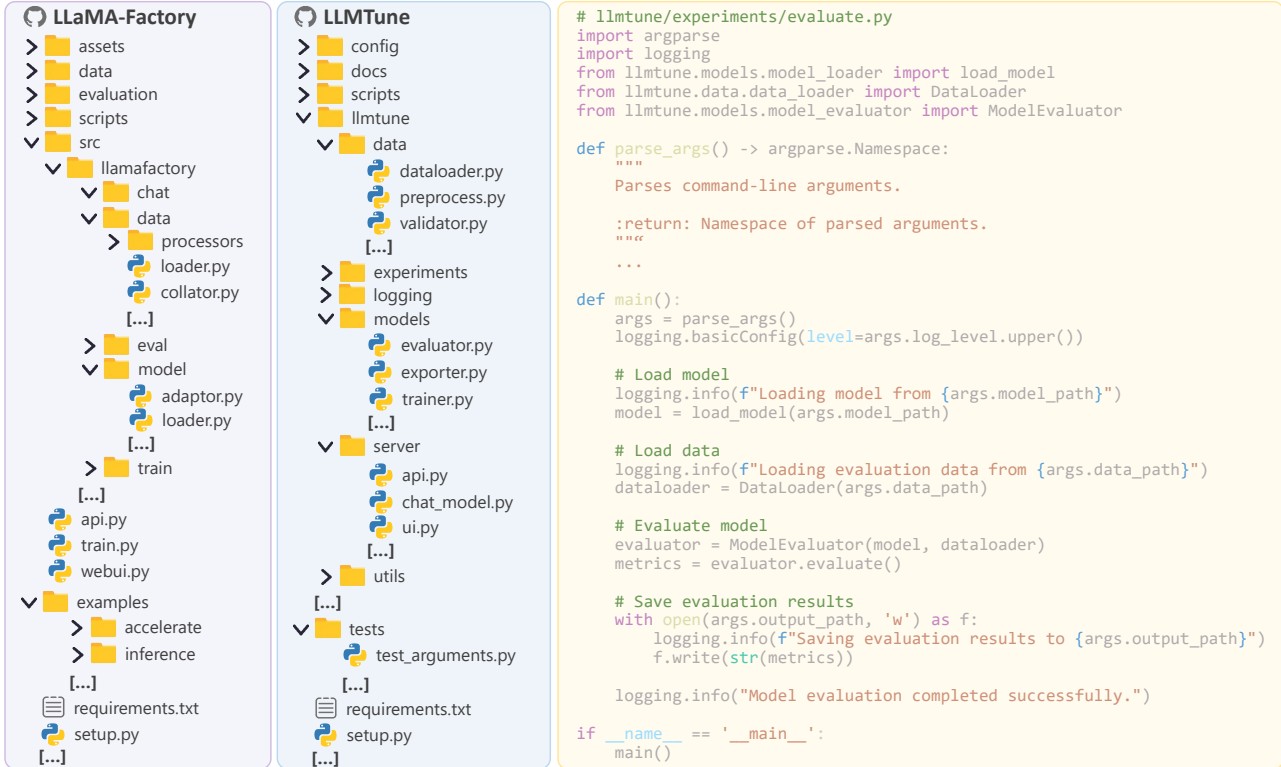

Figure 5: An example of our repo-level code generation. The left part shows the original LLaMA-Factory repository structure, the middle part presents the structure of LLMTune, which we generated based on the extracted feature tree, and the right part illustrates an example file from the generated repository.

we can dynamically control the complexity of the synthetic data, ensuring adaptability to diverse requirements. Code complexity is evaluated through two approaches: 1) using various software engineering principles; 2) leveraging an external LLM judge to evaluate from multiple perspectives.

### 4.1.1. USING SOFTWARE ENGINEERING PRINCIPLES

We first adopt software engineering principles to compare the complexity of our generated code (at both the function and file levels) with existing code datasets. This comparison is based on three well-established software engineering metrics: Halstead Complexity (Halstead, 1977), Strictness Complexity (Ray et al., 2016), and Cyclomatic Complexity (McCabe, 1976). Halstead measures logical complexity via operands and operators, Strictness evaluates execution path strictness, and Cyclomatic quantifies control flow complexity through branching analysis.

Table 2 compares Halstead complexity, with detailed results in Appendix C.3.1. Our function-level data outperforms the runner-up by 2.55 and 20.99 for unique operators and operands, and by 56.98 and 100.36 for total counts, nearly doubling the baseline. File-level results further amplify this trend, significantly exceeding function-level metrics.

Similarly, Table 12 in Appendix C.3.1 demonstrates that our dataset exhibits greater strictness and cyclomatic complexity at both the function and file levels. These analyses collectively demonstrate that feature tree-based code synthesis can create code that is both more complex and more sophisticated than existing methods.

### 4.1.2. EVALUATING CODE COMPLEXITY USING LLMS

We evaluate complexity using GPT-4o as a judge, comparing to existing codebases across four dimensions: Error Handling, Modularity, Data Structure Complexity, and Third-

Table 2: Comparison of Halstead complexity between ours and existing codebase. UO: Unique Operators, UP: Unique Operands, TO: Total Operators, TP: Total Operands

| Dataset | UO | UP | TO | TP |
|---|---|---|---|---|
| Code Alpaca | 4.83 | 8.22 | 10.66 | 15.89 |
| CodeFeedBack | 8.11 | 20.42 | 30.98 | 50.05 |
| Evol CodeAlpaca | 7.94 | 18.97 | 29.91 | 46.70 |
| OSS Instruct | 7.44 | 20.99 | 28.05 | 47.55 |
| Ours (func-level) | 10.66 | 44.32 | 56.98 | 100.36 |
| Ours (file-level) | **11.64** | **72.87** | **100.24** | **179.98** |

Table 3: Distribution of unique features across different datasets.

| Datasets | Workflow | Implementation Style | Functionality | Resource Usage | Computation Operation | Security | User Interaction | Data Processing |
|---|---|---|---|---|---|---|---|---|
| Code Alpaca | 994 | 6 | 393 | 7 | 282 | 8 | 82 | 221 |
| CodeFeedback | 2079 | 6 | 535 | 18 | 689 | 48 | 143 | 895 |
| Evol CodeAlpaca | 2163 | 11 | 591 | 21 | 783 | 60 | 134 | 1401 |
| OSS-Instruct | 2254 | 5 | 669 | 39 | 413 | 49 | 192 | 903 |
| Ours (func-level) | 2422 | 6 | 657 | 37 | 819 | 156 | 363 | 2533 |
| Ours (file-level) | 2475 | 11 | 812 | 43 | 536 | 103 | 800 | 2196 |

| Datasets | File Operation | Error Handling | Logging | Dependency Relations | Algorithm | Data Structures | Implementation Logic | Advanced Techniques | Avg. |
|---|---|---|---|---|---|---|---|---|---|
| Code Alpaca | 11 | 54 | 1 | 43 | 232 | 72 | 67 | 10 | 2.48 |
| CodeFeedback | 39 | 229 | 10 | 121 | 427 | 100 | 49 | 63 | 5.45 |
| Evol CodeAlpaca | 55 | 212 | 15 | 226 | 414 | 130 | 74 | 94 | 6.38 |
| OSS-Instruct | 102 | 211 | 62 | 238 | 150 | 140 | 82 | 26 | 5.54 |
| Ours (func-level) | 203 | 357 | 96 | 305 | 316 | 116 | 40 | 100 | 8.53 |
| Ours (file-level) | 387 | 311 | 218 | 447 | 293 | 140 | 69 | 110 | 8.95 |

Party Dependencies (criteria in Appendix C.3.2). Table 4 summarizes the average scores of 5k samples, consistently showing significant improvements. Our function-level code improves by 32.6% over OSS-Instruct, while file-level performance surpasses it by 52.5%, demonstrating our ability to synthesize more complex code.

Table 4: Comparison of complexity . EH: Error Handling, MD: Modularity, DP: Dependency, DS: Data Structure.

| Dataset | EH | MD | DP | DS | Avg. |
|---|---|---|---|---|---|
| Code Alpaca | 2.04 | 2.10 | 2.09 | 2.38 | 2.15 |
| CodeFeedBack | 2.71 | 3.47 | 2.23 | 3.75 | 3.04 |
| Evol CodeAlpaca | 2.53 | 3.32 | 2.66 | 3.58 | 3.02 |
| OSS Instruct | 2.74 | 3.79 | 2.78 | 3.92 | 3.31 |
| Ours (func-level) | 4.11 | 4.71 | 3.83 | 4.90 | 4.39 |
| Ours (file-level) | 4.23 | 5.94 | 4.62 | 5.41 | 5.05 |

## 4.2. Diversity Evaluation

To assess the feature diversity, we sample 1k instances from ours, CodeFeedback, and other relevant datasets for comparison. Features are extracted using GPT-4o (with the prompt provided in Appendix C.4.1), and the number of unique features in each dataset is reported.

Table 3 shows that our dataset surpasses others in feature diversity, achieving an average of 8.53 unique features at the function level and 8.95 at the file level, outperforming the nearest competitor by 2.15 and 2.57, respectively.

Our function-level data significantly improves in areas like data processing, error handling, dependency relations, and user interaction, all 2-3 times higher than in existing codebases. Additionally, our data also surpasses existing codebases in total feature count, as detailed in Appendix C.4.2.

## 4.3. Comparison under Comparable Data Size

To further investigate the impact of data size and data quality on model performance, we conducted two additional experiments comparing EpiCoder with existing baselines under similar data size conditions.

**Comparison with Magicoder and WaveCoder.** We sampled 75k function-level data from EpiCoder's full dataset and fine-tuned the DeepSeek-Coder-Base-6.7B model, referred to as **EpiCoder-DS-6.7B-Sample75k**. We then compared it with **Magicoder-DS (75k)** and **WaveCoder-Ultra-6.7B (130k)**. As shown in Table 5, despite using the same or fewer data samples, EpiCoder-DS-6.7B-Sample75k achieves higher average performance, demonstrating a **5.4% and 3.0% improvement** over Magicoder-DS and WaveCoder-Ultra-6.7B, respectively.

**Comparison with SelfCodeAlign.** Following the setting in (Wei et al., 2024a), we fine-tuned **CodeQwen-7B** using a 74k subset of EpiCoder function-level data, denoted as **EpiCoder-CodeQwen-7B-Sample74k**. The performance is compared against **SelfCodeAlign-CQ-7B (74k)** in Table 6. EpiCoder-CodeQwen-7B-Sample74k achieves a **4% higher average score**, indicating that the evolutionary data enhancement strategy effectively improves data quality and model generalization.

## 5. Related Work

### 5.1. Code LLMs

Following the success of general LLMs (Brown et al., 2020; Chowdhery et al., 2023) , models like CodeX (Chen et al., 2021) have catalyzed a new surge of research in code intelligence (Sun et al., 2024). The applications of code intelligence have gradually encompass broader real-world scenarios (Wang et al., 2023b; Zhu et al., 2024; Shao et al., 2025). Li et al. (2024) explore the use of LLMs for rewriting

Table 5: Comparison of EpiCoder-DS-6.7B-Sample75k with Magicoder-DS (75k) and WaveCoder-Ultra-6.7B (130k) on function-level benchmarks. All values are Pass@1 (%).

| Model | Data Size | HumanEval | | MBPP | | BCB-Full | | BCB-Hard | | EvoEval | Average |
| | | Base | Plus | Base | Plus | Comp. | Ins. | Comp. | Ins. | | |
|---|---|---|---|---|---|---|---|---|---|---|---|
| Magicoder-DS | 75k | 66.5 | 60.4 | 75.4 | 61.9 | 46.8 | 34.8 | 13.5 | 11.5 | 41.2 | 45.8 |
| WaveCoder-Ultra-6.7B | 130k | 75.0 | 69.5 | 74.9 | 63.5 | 43.7 | 33.9 | 16.9 | 12.8 | 43.6 | 48.2 |
| EpiCoder-DS-6.7B-75k | 75k | 78.0 | 73.2 | 79.4 | 68.8 | 48.2 | 35.6 | 18.4 | 12.8 | 46.2 | 51.2 |

Table 6: Comparison of EpiCoder-CodeQwen-7B-Sample74k with SelfCodeAlign-CQ-7B (74k) on function-level benchmarks. All values represent Pass@1 (%).

| Model | Data Size | EvoEval | MBPP+ | HumanEval+ | Average |
|---|---|---|---|---|---|
| SelfCodeAlign-CQ-7B | 74k | 43.6 | 65.2 | 67.1 | **58.6** |
| **EpiCoder-CodeQwen-74k** | 74k | **47.4** | **67.2** | **73.2** | **62.6** |

code to enhance code search performance, while Koziolek et al. (2024) propose a retrieval-augmented method for controlled code generation. Currently, code LLMs are typically developed through continual pre-training (Roziere et al., 2023) and supervised fine-tuning (Yu et al., 2023b) based on general LLMs. Given that general LLMs have already extensively utilized real-world data during their pre-training phases, the construction of data for post-training stages remains a critical issue that requires urgent attention (Ding et al., 2024).

### 5.2. Data Synthesis for Code

Current research indicates that using LLMs to generate synthetic instruction pairs is an effective strategy to address the scarcity of instruction data (Wang et al., 2023a; Yu et al., 2023a). WizardCoder (Luo et al., 2023) employs the Evol-Instruct method to synthesize more complex and diverse instructional data. Similarly, Magicoder (Wei et al., 2024b) utilizes code snippets to guide LLMs in generating high-quality programming problems and solutions. WaveCoder (Yu et al., 2023b) proposed a generator-discriminator framework based on LLMs to produce diverse and high-quality instruction data. OpenCodeInterpreter (Zheng et al., 2024) constructs datasets through interactions of users, LLMs, and compilers, aiming to meet specific criteria such as diversity, challenge, and multi-turn dialogue structure. Genetic-Instruct (Majumdar et al., 2024) simulates the evolutionary processes of natural selection and genetic algorithms, employing crossover and mutation operations to generate new instructional samples. SelfCodeAlign (Wei et al., 2024a) extracts code concepts to generate new data. These methods collectively demonstrate the efficacy of leveraging LLMs to synthesize instruction data, significantly enhancing the coding capabilities of language models.

### 6. Conclusion

In this work, we introduce a feature tree-based synthesis framework for generating diverse and complex code instruction data. Inspired by Abstract Syntax Trees (AST), our approach constructs hierarchical feature trees to capture semantic relationships within code, enabling scalable synthesis of instruction data with controllable complexity. The experimental results demonstrate that our synthesized data excels in both diversity and complexity, and the trained EpiCoder achieves outstanding performance in tasks of varying complexity, from function-level to file-level benchmarks. Moreover, our approach shows strong potential for scaling to repository-level code synthesis and advancing the usability of code LLMs in complex programming environments.

### Acknowledgment

Yaoxiang Wang and Jinsong Su are supported by National Natural Science Foundation of China (No. 62036004 and No. 62276219), Natural Science Foundation of Fujian Province of China (No. 2024J011001), and the Public Technology Service Platform Project of Xiamen (No.3502Z20231043). Haoling Li, Jie Wu and Yujiu Yang are supported by the Shenzhen Science and Technology Program (JCYJ20220818101001004) and the research grant No. CT20240905126002 of the Doubao Large Model Fund. We also thank the reviewers for their insightful comments.

### Impact Statement

In this work, we propose a feature tree-based code synthesis framework that enables the generation of diverse and complex code instruction data. Our approach improves the performance of code LLMs across tasks of varying complexity, enhancing their applicability in real-world programming scenarios. This, in turn, has the potential to significantly

improve the efficiency of professionals working in software development and related fields.

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

# A. Appendix of Data Synthesis Framework

In this section, we first present an example of a file-level code synthesized by our framework (Section A.1). Then, we provide detailed implementations for several key components of the framework, including **Feature Tree Extraction** (Section A.2), **Feature Tree Evolution** (Section A.3), **Task Generation** (Section A.4), and **Code Generation** (Section A.5).

## A.1. Case of generated File-level Code

This is an example of the file-level code we generated.

Figure 6: An illustrative example of file-level code generation (including the corresponding test code file). Different files contain distinct functional modules, with complex dependencies existing across multiple files.

## A.2. Feature Tree Extraction

Here are our draft prompts for pre-extraction and the refined prompts for feature tree extraction. For brevity, only a portion is shown here. The complete prompts can be found in our released code.

---

**Draft Prompts for Pre-extraction**

Extract high-level information from a code snippet using keywords separated by "##". For example:
**1. Function Description:** Describe the main functionality of the code. Use keywords such as `list sorting ## input parsing ## data storage ## image processing`.
**2. Algorithm Details:** Describe the specific algorithm used and its characteristics. Use keywords such as `dynamic programming ## greedy algorithm ## divide and conquer ## backtracking ## graph traversal`.
**3.** ...
Please use this code as input and extract as much of the specified information as possible based on the content of the code.
**Input:** {source_code}
**Output:** `<your answer>`

---

---

**Algorithm 2** Feature Evolution with Frequency Estimation

---

1: **Input:** Feature tree $T$, frequency library $F$ containing the frequency of each node in $T$, maximum steps **N**
2: **Output:** Updated frequency library $F$
3: **for** step = 1 to **N do**
4:    $t \leftarrow \text{sample}(T)$                                                       {Sample a subtree $t$ from $T$}
5:    $expanded\_t \leftarrow \text{LLM.evolve}(t)$                                 {Evolve $t$ along depth and breadth}
6:    **for** each $node \in expanded\_t \setminus t$ **do**
7:       $S \leftarrow \text{find\_siblings}(node, expanded\_t)$                       {Find siblings of $node$ in $expanded\_t$}
8:       **if** $S = \emptyset$ **then**
9:          $S \leftarrow \text{find\_siblings}(node, T)$           {If no siblings in $expanded\_t$, find siblings in $T$}
10:       **end if**
11:       **if** $S = \emptyset$ **then**
12:          $F(node) \leftarrow 1$                              {Still no siblings found}
13:       **else**
14:          $F(node) \leftarrow \frac{1}{|S|} \sum_{s \in S} F(s)$                        {Update frequency}
15:       **end if**
16:    **end for**
17: **end for**

---

> **Part of Refined Prompts for Feature Tree Extraction**
>
> Extract features from the provided code snippets, following the requirements for each category below, formatted in JSON structure.
> **Categories to extract:**
> **1. Programming Language:** Note the specific programming language used. Example: `["Python", "Java"]`.
> **2. Workflow:** Outline the main steps and operations the code performs. Example: `["data loading", "preprocessing", "model training", "evaluation", "results saving"]`.
> **3. Implementation Style:** What programming paradigm the code follows. Example: `["procedural", "object-oriented", "functional"]`.
> **4. Functionality:** Explain the function of the code. Example: `["data processing", "user interaction", "system control"]`.
> **5. Resource Usage:** Analyze how the code utilizes system resources. Example: `["CPU Cycles", "GPU ComputeOperations", "Network Bandwidth"]`.
> **6. Data Processing:** Describe how the data is processed. This category can include the following subcategories:
>
> - **6.1 Data Preparation:** Steps taken to prepare data for further processing. Example: `["validate units", "strip whitespace"]`.
>
> - **6.2 Data Retrieval:** Methods for obtaining data. Example: `["fetching records", "retrieve top-level items"]`.
>
> - **6.3 Data Transformation:** Describe data transformation operations. Example: `["convert to numpy array", "jsonschema"]`.
>
> - Other relevant subcategories...
>
> **7. Computation Operation:** What computation operations are used. This category can include the following subcategories:
>
> - **7.1 Mathematical Operation:** Specify mathematical computations, such as calculations involving statistics or algebra. Example: `["standard deviation calculation", "compute power flow"]`.
>
> - **7.2 Algorithmic Operation:** Identify algorithm-based operations, such as those in optimization or data sorting. Example: `["simulated annealing", "Best-Fit Decreasing"]`.
>
> - Other relevant subcategories...
>
> **8. More content is omitted here:** The demonstration tree for extracting additional categories has been truncated for brevity. For the full list of categories and detailed instructions, please refer to our code.
> **Input:** {source_code}
> **Output:** `<your answer>`

## A.3. Feature Tree Evolution

Figure 7 presents an example of the feature evolution. In the experiment, after 9000 steps of evolution, the number of features increased from 5000 to 1,40,000. The estimated frequencies of evolved features are calculated as Algorithm 2.

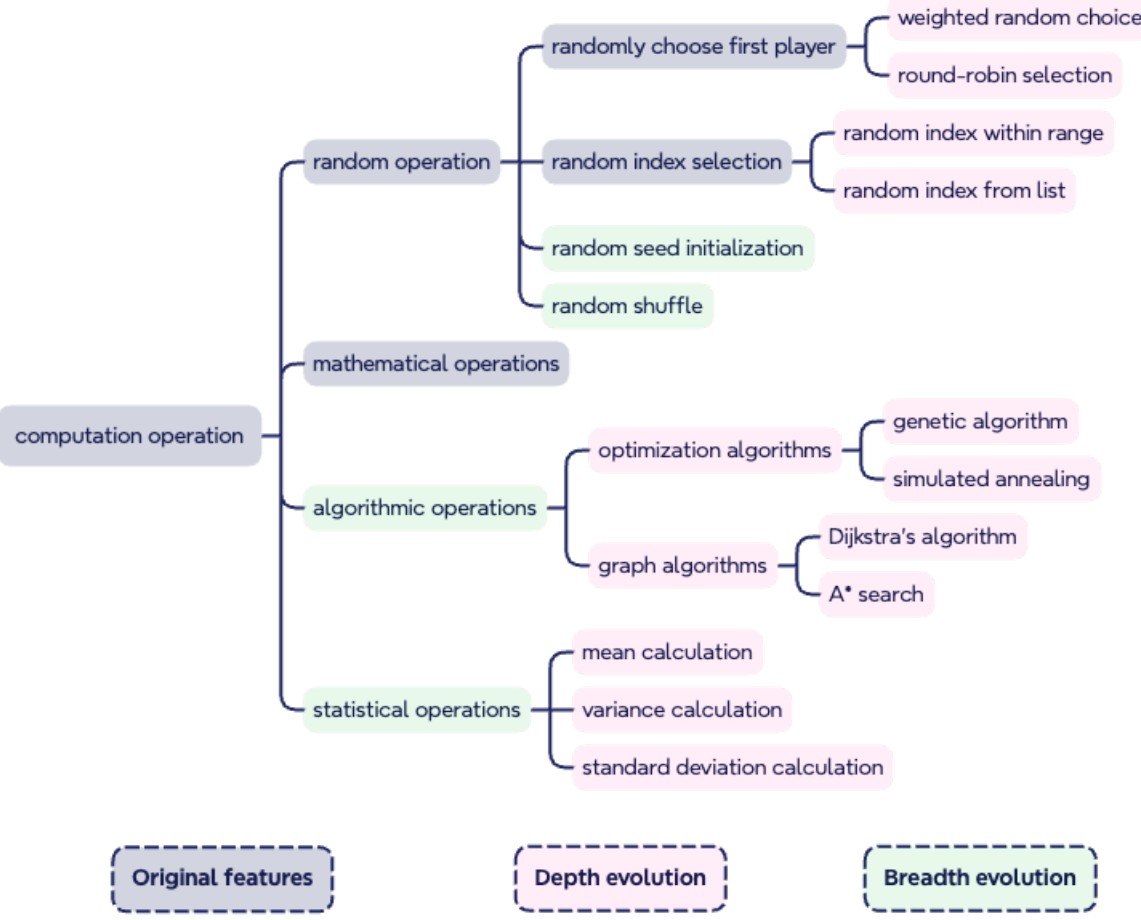

Figure 7: An example of feature evolution.

---

**Prompts for Feature Evolution**

**Feature Tree Evolution Task:** You are provided with a feature tree represented as a nested JSON structure. Each node in this tree represents a feature or a sub-feature of a software system, with the leaves being the most specific features. Your task is to expand this feature tree both in depth and breadth.
Depth expansion means adding more specific sub-features to existing leaves. Breadth expansion means adding more sibling features at the current levels.
Here are some explanations of the features: {explanations}
The input feature tree will be provided in JSON format, and your output should be a JSON structure that represents the expanded feature tree.
**Output Format:**
- Expanded Feature Tree: Provide the expanded feature tree as a JSON structure.
{example}
**Constraints:**
1. For breadth expansion, add at least 2 new sibling features to each existing node.
2. For deep expansion, add new sub-features to any leaf node that could have a more fine-grained feature.
3. Focus on generating new and innovative features that are not present in the provided examples.
Please follow the above constraints and expand the feature tree accordingly.
**Input:** {feature_tree}
**Output:** <begin>expanded feature tree<end>

## A.4. Task Generation

To ensure that the language model (LLM) does not consistently default to familiar or common content, we introduced a strategy to guide the selection of features. From the sampled optional features, certain features are designated as mandatory, and the LLM is directed to incorporate them into the scenario and task. Below is an example of how this approach is applied.

---

### Prompts for Task Generation

You are provided with a set of features/keywords, and a specific programming language. Your task is to use these inputs to conceive a specific real-world application scenario that effectively integrates some of these features. Then, based on the scenario, formulate a task or problem that needs to be addressed with code.

**Procedures:**

1. **Receive Inputs:** These can range from technical specifics like data processing to broader concepts like system interaction.

2. **Select Features:** Choose a combination of features from the provided set that can be realistically integrated into a cohesive scenario.

3. **Conceive a Scenario:** Think of a practical application where the selected features play a critical role.

4. **Formulate a Task Description:** Based on the scenario, formulate a task that needs to be addressed with code. The task should have a certain level of difficulty and test the programmer's coding skills through logical reasoning. Specific details, such as numerical values or environmental conditions, should be included to create a well-defined setting for the task, ensuring the programmer doesn't need to guess any missing information. The task description should not include any code or detailed guidance on code implementation.

5. **Generate an Instruction:** Generate a high-level instruction one or two sentence that describes the overall goal or problem to be solved, without diving into the specific implementation details.

Enclose the selected features with <f> and </f>. Enclose the scenario with  and . Enclose the task with <t> and </t>. Enclose the instruction with  and .

**Inputs:**

- Optional Features: {optional_features}
- Mandatory Features: {mandatory_features}

**Output:**

- **Features:** <f>your answer</f>
- **Scenario:** your answer
- **Task Description:** <t>your answer</t>
- **Instruction:** your answer

---

**Optional Features:**

```
{
    "dependency relations": {
        "time": {
            "time zones handling": []
        },
        "cv2": {
            "cvtColor": []
        }
    },
    "file operation": {
        "read configuration file": [
            "read YAML configuration file"
        ],
        "write data to file": [
            "write to CSV file"
```

```
        ]
    },
    "workflow": {
        "data augmentation": [
            "audio augmentation"
        ],
        "validation": [
            "check data integrity"
        ]
    },
    "programming language": "Python"
}
```

**Mandatory Features:**

```
["read YAML configuration file"]
```

**Selected Features:**

```
["read YAML configuration file, write to CSV file, check data integrity"]
```

**Scenario:** A company wants to automate the generation of a daily status report for its distributed servers. The configuration for each server (such as IP address, location, and operational status) is stored in a YAML file. The system needs to read this configuration file, validate the data to ensure there are no missing or malformed entries, and then write the validated data to a CSV file that can be used for further analysis.

**Task:** Create a function in Python that reads server configuration data from a YAML file, validates the integrity of the data (checking for missing or malformed entries), and writes the validated data to a CSV file. The YAML file will contain a list of servers with their respective details such as IP address, location, and operational status. The function should ensure that all required fields are present and correctly formatted before writing the data to the CSV file. The CSV file should have columns for each server detail and only include entries that have passed validation.

**Instruction:** Develop a Python function that reads, validates, and exports server config data from a YAML to a CSV file.

## A.5. Code Generation

To demonstrate the process of generating code based on the previously outlined task, we present the prompts used, the generated code, its test cases, and the debugging process leading to the final implementation.

### A.5.1. PROMPT FOR CODE GENERATION

The prompt provided specifies a detailed structure for code generation, including the expected outputs and format.

---

**Code Generation Prompt**

Now that you are a code expert, I have provided you with the QUESTION. Complete the problem with awesome code logic and give a richly commented analysis in the code of your answer. Include the necessary packages and test cases.
- **QUESTION**
{task}
- **Full code implementation with test cases**
Enclose the python code with ```python and ``` and enclose the file name with <file> and </file>. For example:
<file>add.py</file>
```python

# add.py
# Code implementation here
def add(x, y):
    return x + y

``` The test code should be in a single file.
<file>test.py</file>
Note that the following code will be executed directly, so only the test cases that can be executed directly need to be retained. You only need to test some simple functions in the code. Tests that depend on external files cannot be executed because these files do not exist.
```python

from add import add

def test():
    assert add(3, 5) == 8
    assert add(4, 6) == 10

test()
```

- **File names in order and packages required**
Answer file names and packages in JSON format, wrapped in <json> and </json> tags. For example:
<json>
{
    "file_names": ["add.py", "test.py"],
    "packages": ["package1", "package2"]
}
</json>

---

## A.5.2. INITIAL CODE IMPLEMENTATION

```python
# server_config.py

import yaml
import csv
import re

def validate_ip(ip):
    """Validate the IP address format"""
    pattern = re.compile(r"^(?:[0-9]{1,3}\.){3}[0-9]{1,3}$")
    return pattern.match(ip) is not None

def validate_server_data(server):
    """Validate the server data"""
    required_fields = ['ip_address', 'location', 'operational_status']

    # Check for missing fields
    for field in required_fields:
        if field not in server:
            return False

    # Validate IP address format
    if not validate_ip(server['ip_address']):
```

```python
            return False

        # Check operational status (assuming it should be a string, non-empty)
        if not isinstance(server['operational_status'], str) or not server['operational_status'].strip():
            return False

        return True

def read_and_validate_yaml(yaml_file):
    """Read and validate the YAML file"""
    with open(yaml_file, 'r') as file:
        try:
            data = yaml.safe_load(file)
            if not isinstance(data, list):
                raise ValueError("YAML file format is incorrect, expected a list of servers.")
            valid_servers = [server for server in data if validate_server_data(server)]
            return valid_servers
        except yaml.YAMLError as exc:
            print(f"Error in YAML file: {exc}")
            return []
        except ValueError as ve:
            print(f"Validation error: {ve}")
            return []

def export_to_csv(servers, csv_file):
    """Export validated server data to CSV"""
    if not servers:
        print("No valid server data to export.")
        return

    keys = servers[0].keys()
    with open(csv_file, 'w', newline='') as file:
        writer = csv.DictWriter(file, fieldnames=keys)
        writer.writeheader()
        writer.writerows(servers)

def process_server_config(yaml_file, csv_file):
    """Main function to process server configuration from YAML to CSV"""
    valid_servers = read_and_validate_yaml(yaml_file)
    export_to_csv(valid_servers, csv_file)
```

File 1: server_config.py

```python
# test_server_config.py

import unittest
import os
from server_config import read_and_validate_yaml, export_to_csv, validate_ip

class TestServerConfig(unittest.TestCase):

    def setUp(self):
        """Set up the test environment"""
        self.valid_yaml_content = """
        - ip_address: 192.168.1.1
          location: Data Center 1
          operational_status: operational
        - ip_address: 10.0.0.2
          location: Data Center 2
          operational_status: maintenance
        """

        self.invalid_yaml_content = """
        - ip_address: 192.168.1.999
          location: Data Center 1
```

```
                operational_status: operational
            - ip_address: 10.0.0.2
              location: Data Center 2
              operational_status:
            """

            self.mixed_yaml_content = """
            - ip_address: 192.168.1.1
              location: Data Center 1
              operational_status: operational
            - ip_address: 10.0.0.999
              location: Data Center 2
              operational_status: maintenance
            """

            with open('valid_servers.yaml', 'w') as file:
                file.write(self.valid_yaml_content)

            with open('invalid_servers.yaml', 'w') as file:
                file.write(self.invalid_yaml_content)

            with open('mixed_servers.yaml', 'w') as file:
                file.write(self.mixed_yaml_content)

    def tearDown(self):
        """Clean up after tests"""
        os.remove('valid_servers.yaml')
        os.remove('invalid_servers.yaml')
        os.remove('mixed_servers.yaml')
        if os.path.exists('output.csv'):
            os.remove('output.csv')

    def test_validate_ip(self):
        """Test IP validation function"""
        self.assertTrue(validate_ip('192.168.1.1'))
        self.assertFalse(validate_ip('192.168.1.999'))
        self.assertFalse(validate_ip('invalid_ip'))

    def test_read_and_validate_yaml(self):
        """Test reading and validating YAML data"""
        valid_servers = read_and_validate_yaml('valid_servers.yaml')
        self.assertEqual(len(valid_servers), 2)

        invalid_servers = read_and_validate_yaml('invalid_servers.yaml')
        self.assertEqual(len(invalid_servers), 0)

        mixed_servers = read_and_validate_yaml('mixed_servers.yaml')
        self.assertEqual(len(mixed_servers), 1)

    def test_export_to_csv(self):
        """Test exporting data to CSV"""
        valid_servers = read_and_validate_yaml('valid_servers.yaml')
        export_to_csv(valid_servers, 'output.csv')

        with open('output.csv', 'r') as file:
            content = file.read()
            self.assertIn('ip_address,location,operational_status', content)
            self.assertIn('192.168.1.1,Data Center 1,operational', content)
            self.assertIn('10.0.0.2,Data Center 2,maintenance', content)

if __name__ == "__main__":
    unittest.main()
```

File 2: test_server_config.py

A.5.3. DEBUGGING AND OPTIMIZATION

Initial execution of the tests revealed several issues:

- The IP validation function incorrectly accepted some invalid IP addresses, such as 192.168.1.999.

- Edge cases in operational status validation were not handled properly.

Given the error message and the original code, the LLM gives a corrected implementation.

```python
# server_config.py

import yaml
import csv
import re

def validate_ip(ip):
    """Validate the IP address format"""
    pattern = re.compile(r"^(?:[0-9]{1,3}\.){3}[0-9]{1,3}$")
    if not pattern.match(ip):
        return False
    parts = ip.split('.')
    for part in parts:
        if not 0 <= int(part) <= 255:
            return False
    return True

# The rest remains unchanged
```

File 3: corrected server_config.py

# B. Appendix of Evaluation

In this section, we introduce the function-level code benches and provide results on EvolEval and FullStackBench (section B.1), then detail the construction of XFileDep (section B.2) and provide a case of file-level code generation (section B.3).

## B.1. Function-level Code Generation Benchmark

We detail the individual function-level code generation benchmarks in this subsection, as well as more detailed results.

**HumanEval and MBPP** HumanEval (Chen et al., 2021) and MBPP (Austin et al., 2021) are popular benchmarks for assessing code generation. Considering the limited test cases in these benchmarks, we followed previous work (Wei et al., 2024b; Zheng et al., 2024) and utilized the EvalPlus (Liu et al., 2023) framework to evaluate model robustness across a broader range of test cases. To ensure fair comparison, we used version 0.2.0 of MBPP+ provided by EvalPlus, which removes some broken tasks (399 → 378 tasks). Table 1 shows the results of different LLMs on these benchmarks.

**BigCodeBench** BigCodeBench (Zhuo et al., 2025) is a comprehensive benchmark designed to assess a model's ability to handle real-world programming tasks, particularly its effectiveness in utilizing various function calls as tools. Our model's ability to adeptly manage these high-complexity scenarios underscores its suitability for BigCodeBench.

**EvoEval** Many benchmarks are prone to data leakage. To mitigate this, we comprehensively evaluate LLM coding capabilities on EvoEval (Xia et al., 2024), constructed by evolving HumanEval to different target domains (Difficult, Creative, Subtle, Combine, and Tool Use). Table 7 shows results of different LLMs. We obtained results from (Xia et al., 2024) and, for models without reported results, conducted tests using their default prompts.

Table 7: Pass@1 (%) results of different LLMs on EvoEval computed with greedy decoding.

| Model | Difficult | Creative | Subtle | Combine | Tool Use | Avg |
|---|---|---|---|---|---|---|
| Closed-source Model | | | | | | |
| GPT-4-Turbo | 50.0 | 61.0 | 82.0 | 45.0 | 69.0 | 61.4 |
| GPT-4 | 52.0 | 66.0 | 76.0 | 53.0 | 68.0 | 63.0 |
| Claude-3 | 50.0 | 53.0 | 81.0 | 42.0 | 69.0 | 59.0 |
| ChatGPT | 33.0 | 42.0 | 70.0 | 33.0 | 64.0 | 48.4 |
| Claude-3-haiku | 40.0 | 47.0 | 65.0 | 17.0 | 56.0 | 45.0 |
| 7B+ Scale | | | | | | |
| Qwen2.5-Coder-32B-Instruct | 57.0 | 58.0 | 90.0 | 58.0 | 75.0 | 67.6 |
| Codestral-22B-v0.1 | 52.0 | 53.0 | 69.0 | 41.0 | 71.0 | 57.2 |
| DeepSeekCoder-33b-Instruct | 47.0 | 47.0 | 67.0 | 31.0 | 66.0 | 51.6 |
| WizardCoder-33b-1.1 | 48.0 | 48.0 | 66.0 | 20.0 | 64.0 | 49.2 |
| CodeLlama-70b-Instruct | 31.0 | 41.0 | 65.0 | 18.0 | 65.0 | 44.0 |
| OpenCoder-8B-Instruct | 45.0 | 50.0 | 73.0 | 28.0 | 50.0 | 49.2 |
| ~ 7B Scale | | | | | | |
| DeepSeek-Coder-6.7B-base | 21.0 | 24.0 | 47.0 | 5.0 | 55.0 | 30.4 |
| DeepSeekCoder-6.7b-Instruct | 40.0 | 37.0 | 61.0 | 18.0 | 51.0 | 41.4 |
| Magicoder-S-DS-6.7B | 40.0 | 34.0 | 67.0 | 21.0 | 61.0 | 44.6 |
| WaveCoder-Ultra-6.7B | 38.0 | 42.0 | 71.0 | 24.0 | 35.0 | 42.0 |
| OpenCodeInterpreter-DS-6.7B | **43.0** | 37.0 | 65.0 | 25.0 | 51.0 | 44.2 |
| **EpiCoder-DS-6.7B** | 40.0 | **45.0** | **70.0** | **30.0** | **65.0** | **50.0** |
| Qwen2.5-Coder-7B-Base | 35.0 | 20.0 | 55.0 | 27.0 | 41.0 | 35.6 |
| Qwen2.5-Coder-7B-Instruct | 48.0 | **49.0** | 77.0 | 37.0 | 65.0 | 55.2 |
| **EpiCoder-Qwen-7B** | **53.0** | 48.0 | **78.0** | **47.0** | **68.0** | **58.8** |

**FullStackBench** Most existing code evaluation datasets cover only limited application areas, such as basic programming and data analysis, lacking a comprehensive and rigorous assessment of code LLMs' capabilities across broader and more complex computer science domains. To convincingly demonstrate our model's strong performance across a wide and diverse range of areas, we utilize FullStackBench (Liu et al., 2024). This benchmark encompasses 16 programming languages and various computer science domains, aiming to thoroughly and systematically evaluate the abilities of large models in diverse real-world coding scenarios. Table 8 shows the results of different LLMs on the FullStackBench.

Table 8: Model performance across domains of Python in the English Subset of FullStackBench.

| Model | BP | AP | SE | DP | MA | DW | ML | SC | DB | MM | OS | Others | Overall |
|---|---|---|---|---|---|---|---|---|---|---|---|---|---|
| Close-Sourced API Model | | | | | | | | | | | | | |
| OpenAI o1-preview | 55.56 | 78.61 | 64.29 | 76.80 | 79.14 | 18.75 | 51.28 | 61.76 | 40.00 | 47.37 | 100.00 | 74.47 | 66.47 |
| OpenAI o1-mini | 72.22 | 75.62 | 50.00 | 76.00 | 80.58 | 28.75 | 56.41 | 56.62 | 40.00 | 57.89 | 100.00 | 72.34 | 66.23 |
| Claude-35-Sonnet | 50.00 | 75.62 | 71.43 | 76.00 | 76.26 | 13.75 | 51.28 | 61.76 | 50.00 | 63.16 | 100.00 | 78.72 | 65.52 |
| GPT 4o-0806 | 72.22 | 72.14 | 53.57 | 78.40 | 76.98 | 21.25 | 66.67 | 55.15 | 40.00 | 68.42 | 100.00 | 72.34 | 65.05 |
| Doubao-Coder-Preview | 55.56 | 69.65 | 50.00 | 77.60 | 75.54 | 27.50 | 51.28 | 60.29 | 20.00 | 63.16 | 50.00 | 55.32 | 62.91 |
| DeepSeek-v2.5 | 55.56 | 68.16 | 50.00 | 76.00 | 76.26 | 20.00 | 48.72 | 56.62 | 40.00 | 63.16 | 50.00 | 65.96 | 61.85 |
| Qwen-Max | 50.00 | 70.15 | 39.29 | 77.60 | 72.66 | 13.75 | 56.41 | 57.35 | 30.00 | 47.37 | 50.00 | 63.83 | 60.78 |
| GLM-4-Plus | 55.56 | 65.67 | 39.29 | 76.80 | 74.82 | 13.75 | 58.97 | 50.00 | 40.00 | 52.63 | 100.00 | 53.19 | 58.77 |
| 20B+ Instruction Tuned Coder | | | | | | | | | | | | | |
| DeepSeekCoder-v2-Instruct | 55.56 | 68.66 | 35.71 | 81.60 | 79.14 | 16.25 | 48.72 | 53.68 | 40.00 | 52.63 | 50.00 | 57.45 | 61.26 |
| Qwen2.5-Coder-32B-Instruct | 50.00 | 70.15 | 50.00 | 77.60 | 66.19 | 17.50 | 61.54 | 43.38 | 30.00 | 47.37 | 100.00 | 61.70 | 58.41 |
| DeepSeekCoder-33B-Instruct | 50.00 | 59.70 | 21.43 | 71.20 | 48.92 | 18.75 | 48.72 | 40.44 | 30.00 | 42.11 | 50.00 | 44.68 | 49.05 |
| CodeLlama-34B-Instruct | 5.56 | 22.89 | 14.29 | 40.00 | 17.27 | 16.25 | 15.38 | 18.38 | 30.00 | 26.32 | 0.00 | 23.40 | 22.27 |
| 13B+ Instruction Tuned Coder | | | | | | | | | | | | | |
| Qwen2.5-Coder-14B-Instruct | 55.56 | 62.69 | 32.14 | 76.00 | 70.50 | 18.75 | 53.85 | 38.97 | 30.00 | 57.89 | 100.00 | 55.32 | 55.57 |
| DeepSeekCoder-v2-Lite-Instruct | 50.00 | 64.68 | 32.14 | 64.00 | 56.12 | 26.25 | 43.59 | 33.82 | 60.00 | 21.05 | 50.00 | 53.19 | 50.47 |
| StarCoder2-15B-Instruct-v0.1 | 61.11 | 44.28 | 32.14 | 63.20 | 36.69 | 31.25 | 53.85 | 28.68 | 60.00 | 36.84 | 50.00 | 53.19 | 43.01 |
| CodeLlama-13B-Instruct | 11.11 | 22.39 | 25.00 | 24.00 | 20.86 | 30.00 | 20.51 | 13.97 | 40.00 | 10.53 | 50.00 | 23.40 | 21.56 |
| 6B+ Instruction Tuned Coder | | | | | | | | | | | | | |
| Qwen2.5-Coder-7B-Instruct | 33.33 | 58.21 | 39.29 | 66.40 | 48.92 | 18.75 | 38.46 | 32.35 | 40.00 | 47.37 | 50.00 | 59.57 | 47.51 |
| Yi-Coder-9B-Chat | 61.11 | 50.25 | 32.14 | 66.40 | 46.76 | 26.25 | 43.59 | 36.76 | 50.00 | 36.84 | 50.00 | 48.94 | 46.56 |
| DeepSeek-Coder-7B-Instruct-v1.5 | 50.00 | 51.74 | 25.00 | 64.80 | 37.41 | 25.00 | 30.77 | 34.56 | 20.00 | 52.63 | 50.00 | 48.94 | 43.60 |
| OpenCoder-8B-Instruct | 44.44 | 53.73 | 28.57 | 57.60 | 35.97 | 26.25 | 28.21 | 28.68 | 0.00 | 47.37 | 0.00 | 44.68 | 41.11 |
| DeepSeek-Coder-6.7B-Instruct | 61.11 | 49.75 | 28.57 | 65.60 | 38.13 | 18.75 | 38.46 | 22.79 | 30.00 | 31.58 | 50.00 | 42.55 | 40.88 |
| CodeQwen1.5-7B-Chat | 38.89 | 45.77 | 50.00 | 58.40 | 31.65 | 15.00 | 33.33 | 22.79 | 20.00 | 31.58 | 0.00 | 42.55 | 37.20 |
| CodeLlama-7B-Instruct | 27.78 | 23.88 | 25.00 | 28.00 | 20.86 | 23.75 | 10.26 | 11.76 | 50.00 | 10.53 | 0.00 | 21.28 | 21.33 |
| **EpiCoder-DS-6.7B** | **61.11** | **47.26** | **25.00** | **61.60** | **41.01** | **40.00** | **41.03** | **27.21** | **50.00** | **36.84** | **50.00** | **42.55** | **43.25** |
| **EpiCoder-Qwen-7B** | **44.44** | **61.19** | **17.86** | **72.80** | **61.15** | **28.75** | **51.28** | **27.94** | **20.00** | **47.37** | **50.00** | **40.43** | **50.24** |

## B.2. Cross-File Dependency Benchmark

The Cross-File Dependency Benchmark (XFileDep) is a specialized benchmark designed to evaluate the performance of code generation models in handling cross-file dependencies. In real-world programming scenarios, there exists a complex web of dependencies between different code files. XFileDep provides a comprehensive framework that tests a model's ability to generate missing files by considering multiple interdependent files as context. This benchmark not only measures the model's capability to generate individual isolated files but also assesses its proficiency in understanding and managing dependencies between files. As illustrated in Figure 8 , the constructing the XFileDep consists of following steps.

**Step 1: Data Sample Selection.** The construction of the XFileDep starts from carefully data selection. From the initial cross-file dataset of 35,000 data samples constructed using our pipeline of synthetic data based on extracted features, we carefully filtered out cross-file data samples with fewer than 5 interconnected files (excluding any test files), resulting in a refined set of 3,435 high-quality samples. Figure 9 comprehensively displays the distribution of file counts, the average file length for each data sample, and the overall structural characteristics of the dataset.

**Step 2: Dependency File Selection.** We utilized Abstract Syntax Trees (AST) to conduct dependency analysis and structural identification of the Python code files. AST allows parsing of the syntactic structure of Python files, enabling extraction of module import dependencies, function definitions, and class definitions. By parsing all code files within the data sample and identifying the collaboration between classes and functions, we documented the details of defined functions and classes along with the information on imported modules. With these capabilities, we were able to traverse the entire data sample, analyze the dependency relationships between files, discern key files, and select a representative candidate file (to serve as the target file for the code generation task) that has a substantial amount of cross-file dependencies. This approach allows us to generate a structured data analysis report. The systematic nature of this analysis allows us to efficiently handle large cross-file data, providing clear dependency graphs and detailed information on code structure. We also filtered out data samples that lacked rich cross-file dependencies, retaining 2,934 samples that met the criteria.

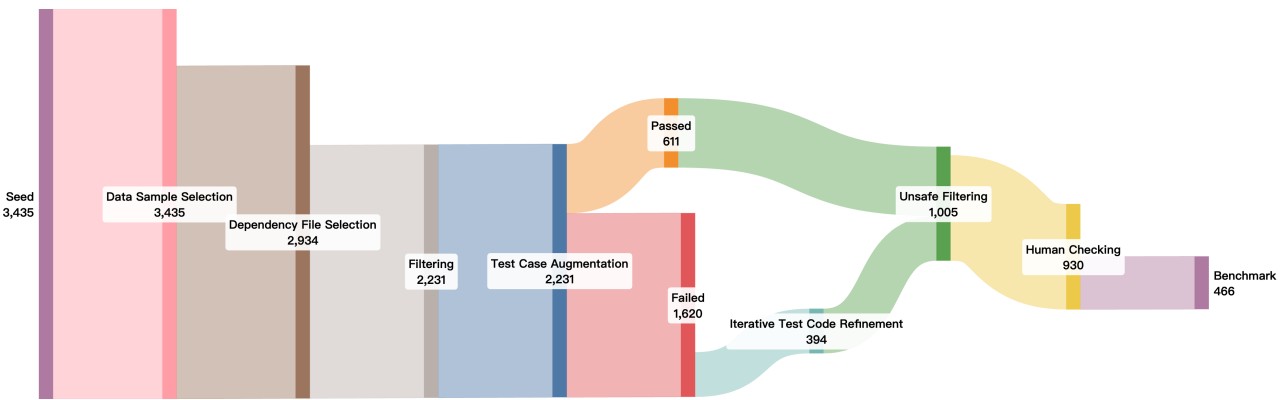

Figure 8: The Sankey diagram for XFileDep benchmark creation, with numbers indicating the quantity of data samples.

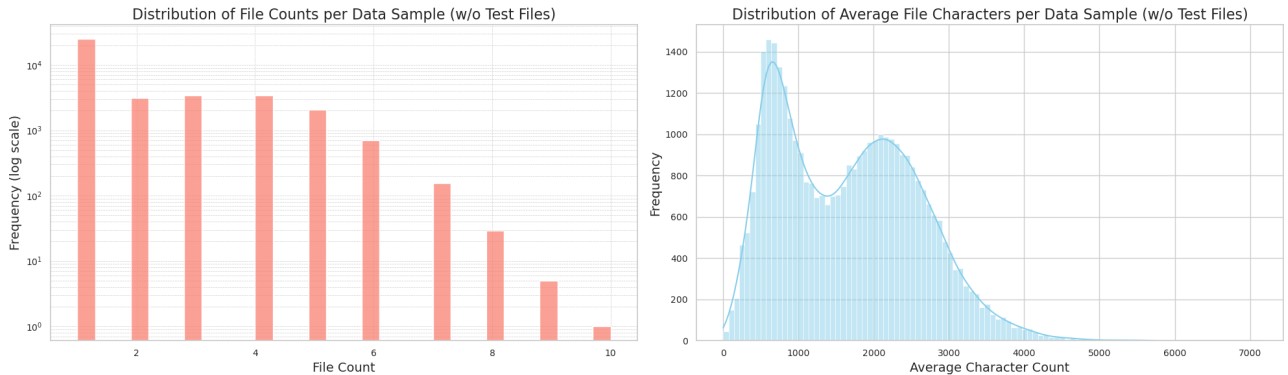

Figure 9: the distribution of file quantities and the average file length for each data sample.

**Step 3: Filtering.** We analyzed the runtime environment, required libraries, and external files (such as .npy, .xlsx, .json, .log) for each data sample. Based on this analysis, data samples that could not provide the necessary files or dependencies were filtered out. We also excluded data samples that had long runtimes or required the generation of specific output files, which made obtaining test results difficult. In addition, to increase the overall difficulty of the task and ensure that cross-file code generation operates at an appropriate file-level length, we filtered out candidate files whose length was less than 300 characters. Finally, we obtained a total of 2,231 data samples.

**Step 4: Test Case Augmentation.** To enhance the test coverage of the code within each data sample, we utilized GPT-4o to generate additional test cases. This process ensured that the core functional methods were robustly and comprehensively tested. In Table 9, we have compiled the statistical data before and after the augmentation of the test cases. We conducted a

Table 9: Comparison of Test Functions and Test Cases before and after augmentation for 930 data samples.

|  | Original | Augmented |
|---|---|---|
| **Test Functions** | | |
| Total | 3,837 | 7,933 |
| Average per sample | 4.13 | 8.53 |
| Max in file | 12 | 23 |
| **Test Cases** | | |
| Total | 6,010 | 14,305 |
| Average per sample | 6.46 | 15.38 |
| Max in file | 20 | 44 |

runtime check on the data after augmenting the test cases and obtained 611 samples that successfully passed the test cases. The prompt for augmenting the test case with GPT-4o is illustrated below.

---

**Prompts for Augmenting Test Cases**

You are an expert in Python programming and test-driven development. I have a repository of Python code with corresponding test files aimed at verifying the correctness of my code. However, I believe the current test cases are insufficient and I need more comprehensive and robust test cases.

**Requirements:**

- The test cases should be written using a suitable testing framework (e.g., pytest, unittest).
- Maintain consistency with existing test structure and naming conventions.
- Ensure that all new test cases pass before finalizing.

**Inputs:**
- The file structure (including filenames) and contents of the Python code.
{python_code}

- The file structure (including filenames) and contents of the current test files.
{test_code}

**Deliverables:**
- Updated test files with additional test cases.
- Documentation or comments within the test files explaining each new test case.

**Your output:**
Returns code content only.

---

**Prompts for Refining Test Code**

You are an expert in Python programming and test-driven development. I have a repository of Python code with corresponding test files aimed at verifying the correctness of my code. However, there are errors in the current test cases and test code that I would like you to proofread and correct.

**Requirements:**
- The test cases should be written using a suitable testing framework (e.g., pytest, unittest).
- Maintain consistency with existing test structure and naming conventions.
- Ensure that all new test cases pass before finalizing.

**Inputs:**
- The file structure (including filenames) and contents of the Python code:
{python_code}

- The file structure (including filenames) and contents of the current test files:
{test_code}

- Program error messages caused by running the test file:
{error_messages}

**Deliverables:**
- Fully correct test code file.
- Documentation or comments within the test files explaining each new test case.
**Your output:**
Returns code content only.

---

**Step 5: Iterative Test Code Refinement.** For data samples that fail the test cases, the code content, test cases, and error information are extracted. Based on these detailed input descriptions, we utilize GPT-4o for checking and modification, and subsequently re-run the refined test code for validation. We performed a single iteration of modification on the test code, resulting in 394 successful test cases out of 1620 samples. Finally we have a sample of 1,005 that pass the test cases. The prompt for refining the test code with GPT-4o is shown above.

**Step 6: Unsafe Filtering.** To ensure the validity of our test cases, we constructed a unit test environment based on the dependency requirements specified in each Python file. We then executed all test cases and filtered out any samples that failed the tests or presented unsafe operations, such as `kill`, `terminate`, `rmtree`, `rmdir`, and `rm`. This approach ensures that our canonical solution is absolutely correct. Finally, we retained 930 samples of cross-file data.

**Step 7: Manual Checking and Verification** To ensure the quality of the questions in the XFileDep, we manually review and verify each question. The evaluation criteria require that the Python code in the `Answer` accurately reflect the functionalities described in the question and produce correct outputs that meet the expected requirements. Any questions that do not meet the criteria will be filtered. All manual reviews are conducted by individuals with at least 5 years of Python programming experience and a master's degree or higher. Additionally, these reviewers are currently employed as software engineers or in similar roles at leading internet companies. After manual review and filtering, we obtained 466 questions.

**Step 8: Annotation.** We selected target files with extensive cross-file dependencies (either frequently invoked by other files or frequently invoking other files). Using GPT-4o, we meticulously annotated all classes and methods in these files with detailed documentation, emphasizing their purpose, functionality, and relationships with other components. The annotation process did not alter the original code in any way, and the successful execution of the annotated files verified the correctness of the ground truth. The full prompt for annotating the target file with GPT-4o is illustrated below.

**Step 9: Benchmark Construction.** To maintain a high level of difficulty in the benchmark construction, we extracted all code blocks from the functions and classes within the target files, leaving only the `import statements`, `FunctionDef`, `ClassDef`, and the corresponding `docstrings`. The instruction set provided the names and contents of all other files in the cross-file data sample as context and included the target file's name and skeleton structure for the completion task.

---

**Prompts for Annotating Target File**

Your task is to read through the provided Python code and add detailed docstrings that describe the purpose and functionality of each class and function. Your additions should follow the PEP 257 conventions and should not alter the original code in any way. The docstrings should provide enough detail to help other developers understand what each part of the code does and how to use it appropriately.

Here is the Python code:

```python
{target_file_code}
```

Please add the necessary docstrings without changing the actual code. Ensure that output is enclosed with its corresponding tags:

```python
[Your code here]
```

---

## B.3. Case of File-Level Code Generation

This section provides comprehensive details on file-level code generation using a generated case. The directory structure of the example and the detailed contents of each file are as follows:

```
--example_root/

    main.py
    optimizer.py
    parser.py
    scraper.py
    search.py
    storage.py

    --tests/
        test_main.py
```

```python
# main.py
from scraper import Scraper
from parser import Parser
from storage import Storage
from search import Search
from optimizer import Optimizer

def main():
    urls = [
        'https://example.com/products',
        # Add more URLs as needed
    ]

    # Step 1: Scrape Data
    scraper = Scraper(urls)
    html_data = scraper.fetch_data()

    # Step 2: Parse Data
    parser = Parser()
    parsed_data = parser.parse(html_data)

    # Step 3: Store Data
    storage = Storage()
    storage.store_data(parsed_data)

    # Optional: Save to JSON
    storage.save_to_json(parsed_data)

    # Step 4: Optimize Code (Example usage)
    code = """
def example_function():
    result = 2 + 2
    return result"""
    optimized_code = Optimizer.optimize_code(code)

    # Step 5: Search Data
    data = storage.fetch_data()
    keyword = 'example'
    search = Search()
    results = search.breadth_first_search(data, keyword)
    print(f"Search results: {results}")

if __name__ == '__main__':
    main()
```

File 4: main.py

```python
# optimizer.py
```

```python
import ast
import compileall

class Optimizer:
    # Example of constant folding optimization
    @staticmethod
    def constant_folding(code):
        tree = ast.parse(code)
        optimized_tree = ast.fix_missing_locations(tree)
        return compile(optimized_tree, filename="<ast>", mode="exec")

    @staticmethod
    def optimize_code(code):
        optimized_code = Optimizer.constant_folding(code)
        compileall.compile_code(optimized_code)
        return optimized_code
```

File 5: optimizer.py

```python
# parser.py
from bs4 import BeautifulSoup

class Parser:
    @staticmethod
    def parse(html_data):
        parsed_data = []
        for html in html_data:
            soup = BeautifulSoup(html, 'html.parser')
            products = []
            for product in soup.select('.product'):
                name = product.select_one('.product-name').text.strip()
                price = product.select_one('.product-price').text.strip()
                description = product.select_one('.product-description').text.strip()
                products.append({
                    'name': name,
                    'price': price,
                    'description': description
                })
            parsed_data.append(products)
        return parsed_data
```

File 6: parser.py

```python
# scraper.py
import requests
from bs4 import BeautifulSoup

class Scraper:
    def __init__(self, urls):
        self.urls = urls

    def fetch_data(self):
        html_data = []
        for url in self.urls:
            try:
                response = requests.get(url)
                response.raise_for_status()
                html_data.append(response.text)
            except requests.RequestException as e:
                print(f"Error fetching data from {url}: {e}")
        return html_data
```

File 7: scraper.py

```python
# search.py
from collections import deque

class Search:
    @staticmethod
    def breadth_first_search(data, keyword):
        queue = deque(data)
        results = []
        while queue:
            item = queue.popleft()
            if keyword.lower() in item['name'].lower() or keyword.lower() in item['description'].lower():
                results.append(item)
        return results
```

File 8: search.py

```python
# storage.py
import json
import sqlite3

class Storage:
    def __init__(self, db_name='data.db'):
        self.conn = sqlite3.connect(db_name)
        self.create_table()

    def create_table(self):
        with self.conn:
            self.conn.execute('''
                CREATE TABLE IF NOT EXISTS products (
                    id INTEGER PRIMARY KEY,
                    name TEXT,
                    price TEXT,
                    description TEXT
                )''')

    def store_data(self, data):
        with self.conn:
            for products in data:
                for product in products:
                    self.conn.execute('''
                        INSERT INTO products (name, price, description)
                        VALUES (:name, :price, :description)''', product)

    def fetch_data(self):
        cursor = self.conn.cursor()
        cursor.execute('SELECT name, price, description FROM products')
        return cursor.fetchall()

    def save_to_json(self, data, filename='data.json'):
        with open(filename, 'w') as f:
            json.dump(data, f, indent=4)
```

File 9: storage.py

```python
# tests/test_main.py
import unittest
from scraper import Scraper
from parser import Parser
from storage import Storage
from search import Search
from unittest.mock import patch

class TestWebScrapingApp(unittest.TestCase):
    @patch('requests.get')
    def test_scraper(self, mock_get):
```

```python
            mock_get.return_value.status_code = 200
            mock_get.return_value.text = '<html><div class="product">Test
                Product$10Test
                Description</div></html>'

            scraper = Scraper(['https://example.com/products'])
            html_data = scraper.fetch_data()
            self.assertEqual(len(html_data), 1)

    def test_parser(self):
        html_data = ['<html><div class="product">Test Product$10Test
            Description</div></html>']
        parser = Parser()
        parsed_data = parser.parse(html_data)
        self.assertEqual(len(parsed_data), 1)
        self.assertEqual(parsed_data[0][0]['name'], 'Test Product')

    def test_storage(self):
        storage = Storage(':memory:')
        data = [[{'name': 'Test Product', 'price': '$10', 'description': 'Test Description'}]]
        storage.store_data(data)
        fetched_data = storage.fetch_data()
        self.assertEqual(len(fetched_data), 1)

    def test_search(self):
        data = [
            {'name': 'Test Product', 'price': '$10', 'description': 'Test Description'},
            {'name': 'Another Product', 'price': '$20', 'description': 'Another Description'}
        ]
        search = Search()
        results = search.breadth_first_search(data, 'Test')
        self.assertEqual(len(results), 1)
        self.assertEqual(results[0]['name'], 'Test Product')

if __name__ == '__main__':
    unittest.main()
```

File 10: tests/test_main.py

# C. Appendix of Analysis

In this section, we first present the scaling effect of code instruction data (section C.1), then discuss the data leakage issue (section C.2), and provide detail analysis regarding complexity (section C.3) and diversity evaluation (section C.4).

### C.1. Scaling of Code Instruction Data

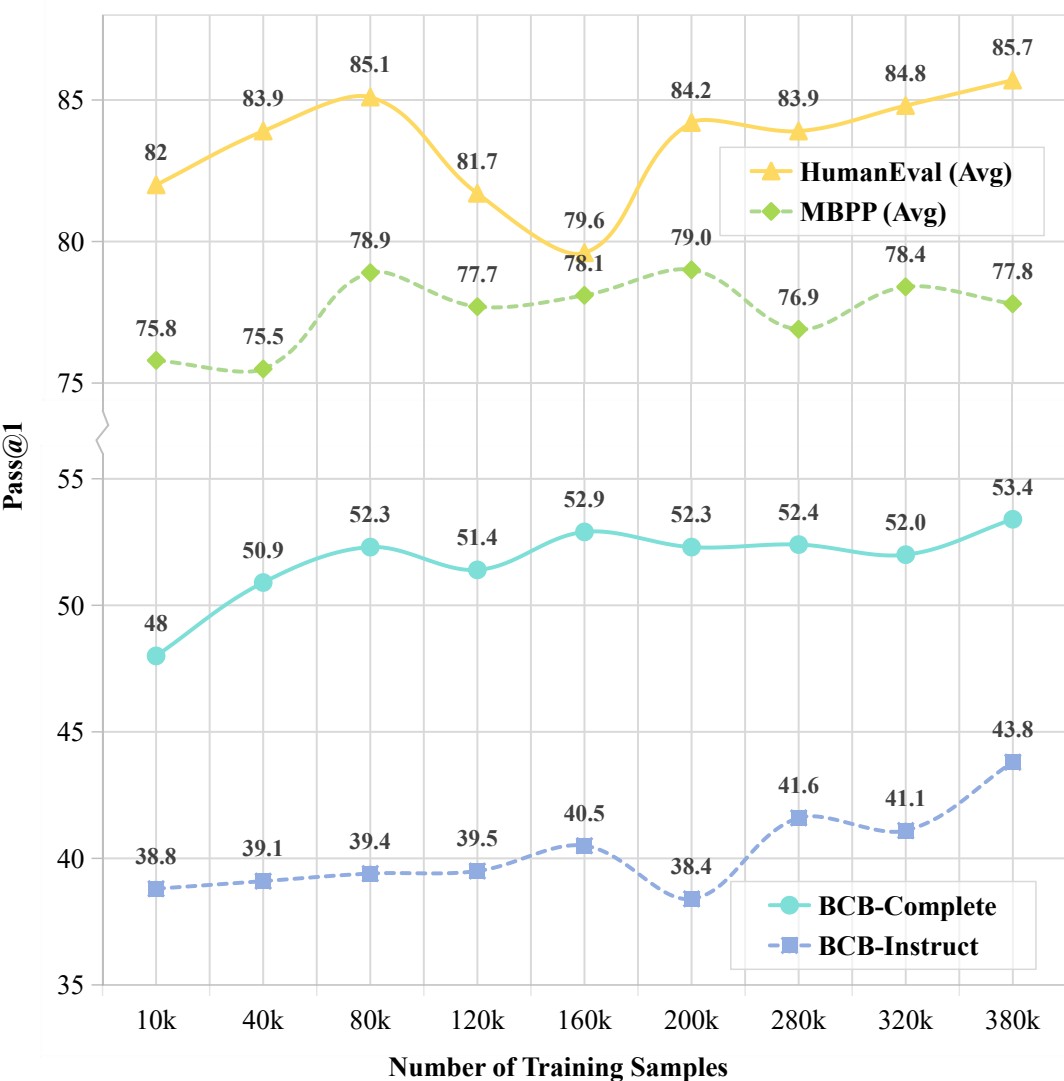

Figure 10: The scaling law of code instruction data. The results obtained from randomly sampled subsets of 380k data points across the HumanEval, MBPP, and BigCodeBench benchmarks.

Although both the fields of mathematics and code are characterized by rigorous logic and precision, they exhibit different phenomena in terms of the quantity of instruction data. Motivated by previous analyses of instruction data scaling laws in the mathematical domain, we design experiments to understand the scaling laws in the code domain. We randomly sample 380k data points and set a series of data quantities for our experiments. The results on the HumanEval, MBPP, and BigCodeBench benchmarks are depicted in Figure 10. It is evident that with the increase in data volume, the performance improves significantly. Moreover, even with the data size reaching 380k, there is still an upward trend, demonstrating that our dataset possesses sufficient diversity to effectively combat overfitting.

## C.2. Data Leakage

### C.2.1. OVERALL RESULTS

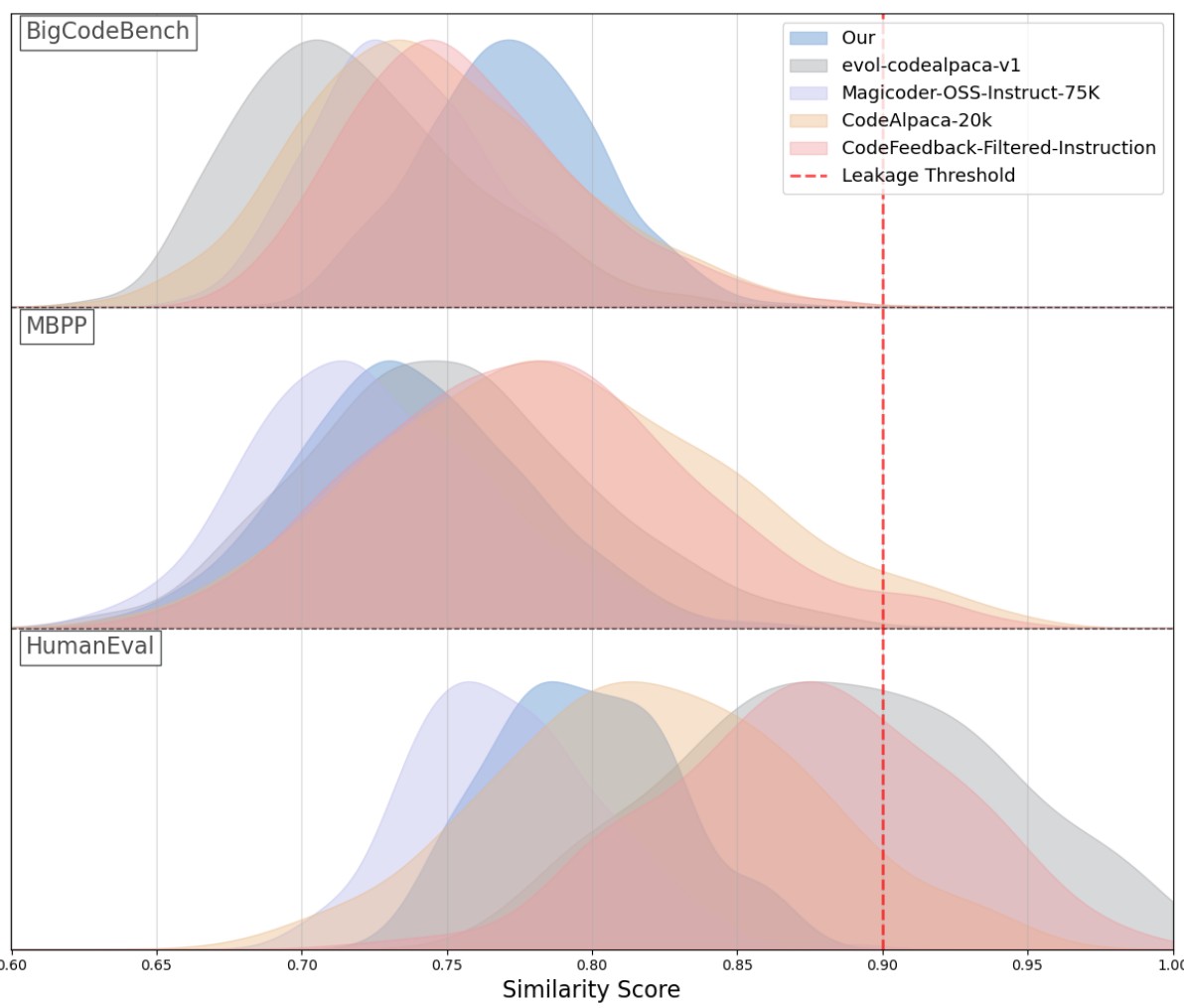

Figure 11: The distribution of cosine similarity scores between various benchmarks HumanEval, MBPP, and BigCodeBench.

We investigate potential data leakage issues to ensure that our synthetic data are free from such risks. Specifically, we use the `jinaai/jina-embeddings-v3` embedding model to generate embeddings for the output segments of all training data, including our synthetic data and other training datasets used for comparison. For the HumanEval, MBPP, and BigCodeBench benchmarks, we encode their test datasets and compute the cosine similarity between each test instance and all training samples. For each test instance in the benchmarks, we identify the training-test data pair with the highest similarity score and plot the distribution of these scores in Figure 11. Furthermore, through a case-based analysis of similarity scores, we define a threshold for potential leakage (`Similarity Score=0.9`), with detailed explanations provided in Appendix C.2.2. Despite the large scale of our dataset, which puts it at a disadvantage when identifying the most similar sample for each test instance, Figure 11 demonstrates that our 380k synthetic function-level data show minimal evidence of data leakage, even for the HumanEval benchmark, where the risk of leakage is most pronounced. Further analysis of similarity scores across other benchmarks supports the conclusion that our synthetic data are not strongly similar to the benchmark. This confirms that our model's performance gains are not due to overfitting to benchmarks but stem from data quality and diversity.

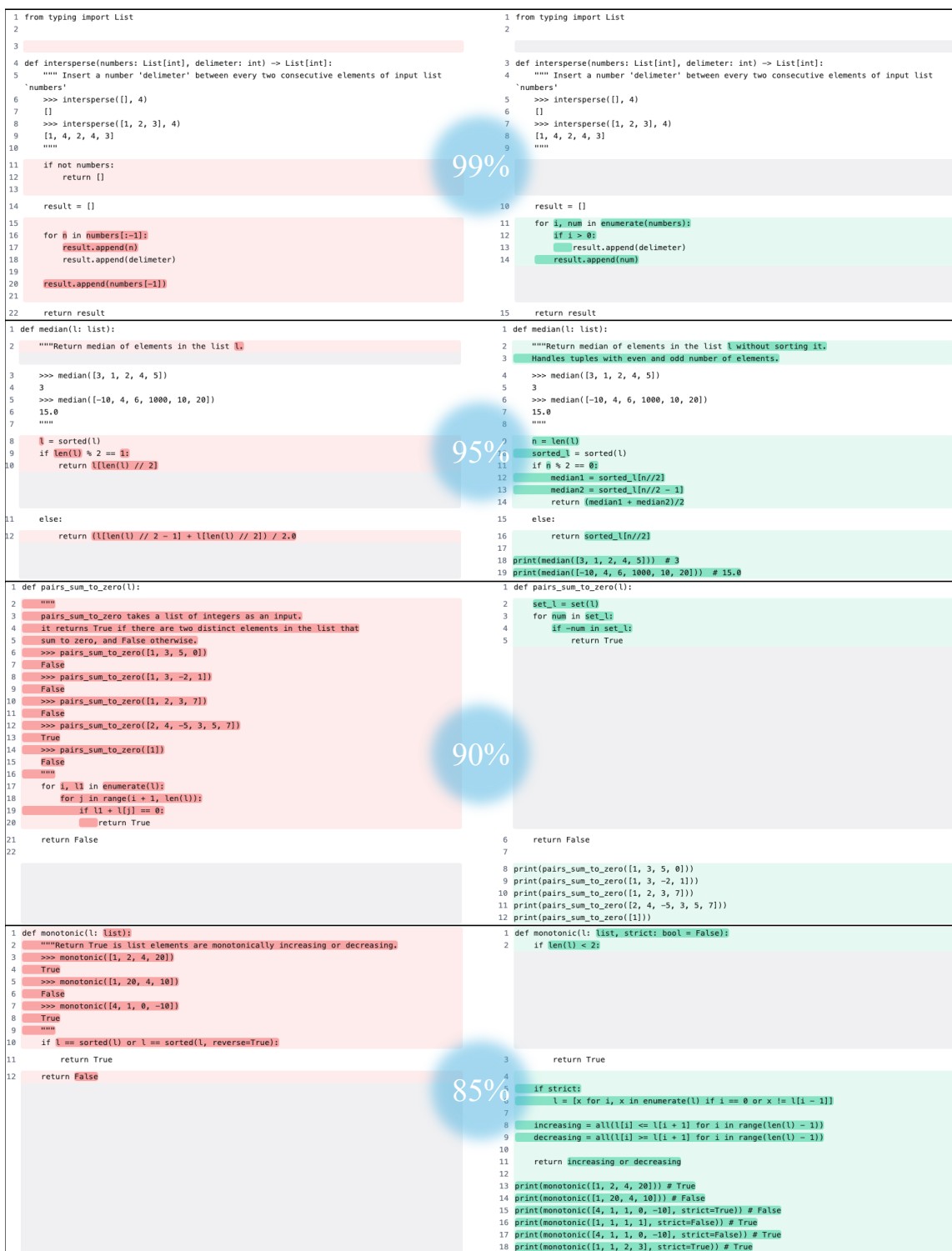

Figure 12: Cases from HumanEval (left) and `evol-codealpaca-v1` (right) with varying similarity. Embeddings are computed based on the "`output`" portions of the trainset and the "`prompt + canonical_solution`" from HumanEval data.

### C.2.2. LEAKAGE THRESHOLD SETTING

We embedded multiple training datasets and different benchmark datasets and calculated the cosine similarity between them. Additionally, we analyzed the most similar training data sample for each test data point to identify potential data leakage issues. Using the `evol-codealpaca-v1` dataset, which exhibited the most severe leakage, as a case study, we found that the training data contained extremely serious data leakage. Figure 12 presents pairs of benchmark data (left) and training data (right) with various similarity scores. The index of the data samples presented in the case study is provided in Table 10. After observing and analyzing multiple samples, we manually selected a similarity score of 0.9 as the similarity threshold.

Table 10: The index of the data samples presented in the case study.

| Similarity | 99% | 95% | 90% | 85% |
|---|---|---|---|---|
| HumanEval index | 5 | 47 | 43 | 57 |
| evol-codealpaca-v1 index | 81260 | 45682 | 9508 | 51167 |

## C.3. Complexity Evaluation

### C.3.1. DETAILED METRICS FROM THE SOFTWARE ENGINEERING PERSPECTIVE.

**Halstead-derived Principle.** We supplement Halstead-derived metrics in Table 11, which are based on unique operators ($n_1$), unique operands ($n_2$), total operators ($N_1$), and total operands ($N_2$). Among these recognized metrics, our data consistently shows significant performance gains compared to the current codebase. For instance, we achieve notable complexity advantages in program volume and program difficulty. The programming effort and estimated programming time further confirm that our data requires more time and effort to achieve. While the increased complexity may suggest a higher potential for bugs, we address this issue by incorporating test cases during the data sythesis.

**Strictness and Cyclomatic Complexity Measures.** Table 12 in demonstrates that our dataset exhibits greater strictness and cyclomatic complexity at both the function and file levels. In Table 13, we break down cyclomatic complexity to observe scores for specific operations, such as *while*, *for*, and *boolean* operations. Table 13 shows that our gains in Cyclomatic complexity are mainly due to the higher occurrence of *if*, *for*, *except*, and *return* statements. This suggests that our program handles more loops and incorporates a broader range of exception handling scenarios. We break down the code strictness complexity scores. Table 14 shows that our data achieves a significantly improvement in *Doc Strings*, indicating a more

Table 11: Derived Halstead metrics. These metrics are derived from unique operators ($n_1$), unique operands ($n_2$), total operators ($N_1$), and total operands ($N_2$).

| Dataset | Program Length (N) $N = N_1 + N_2$ | Vocabulary (n) $n = n_1 + n_2$ | Volume (V) $V = N \times \log_2(n)$ | Difficulty (D) $D = \frac{n_1}{2} \times \frac{N_2}{n_2}$ |
|---|---|---|---|---|
| Code Alpaca | 26.55 | 13.05 | 108.39 | 5.07 |
| Evol CodeAlpaca | 76.61 | 26.91 | 381.45 | 10.76 |
| CodeFeedBack | 81.03 | 28.54 | 416.78 | 10.49 |
| OSS Instruct | 75.61 | 28.43 | 381.32 | 8.75 |
| Ours (func-level) | 157.34 | 54.97 | 932.78 | 12.34 |
| Ours (file-level) | **280.22** | **84.51** | **2035.63** | **13.64** |

| Dataset | Programming Effort (E) $E = D \times V$ | Estimated Time (T) $T = \frac{E}{18}$ | Predicted Bugs (B) $B = \frac{V}{3000}$ |
|---|---|---|---|
| Code Alpaca | 1043.26 | 57.96 | 0.03 |
| Evol CodeAlpaca | 5954.64 | 330.81 | 0.09 |
| CodeFeedBack | 6204.38 | 344.69 | 0.09 |
| OSS Instruct | 4528.98 | 251.61 | 0.08 |
| Ours (func-level) | 13396.28 | 744.24 | 0.17 |
| Ours (file-level) | **67851.94** | **3769.55** | **0.28** |

Table 12: Comparison of Strictness complexity (left) and Cyclomatic complexity (right).

| Dataset | Mean | Median | Std |
|---|---|---|---|
| Code Alpaca | 0.18 | 0.00 | **0.52** |
| Evol CodeAlpaca | 0.82 | 0.00 | 1.63 |
| CodeFeedBack | 0.97 | 0.00 | 2.09 |
| OSS Instruct | 1.50 | 1.00 | 2.19 |
| Ours (func-level) | 4.95 | **4.00** | 3.77 |
| Ours (file-level) | **5.41** | **4.00** | 3.85 |

| Dataset | Mean | Median | Std |
|---|---|---|---|
| Code Alpaca | 2.10 | 1.00 | **1.66** |
| Evol CodeAlpaca | 3.76 | 3.00 | 3.48 |
| CodeFeedBack | 3.96 | 3.00 | 3.33 |
| OSS Instruct | 3.45 | 3.00 | 2.98 |
| Ours (func-level) | 5.14 | 5.00 | 3.01 |
| Ours (file-level) | **14.93** | **14.00** | 6.73 |

Table 13: Comparison of different control flow and logical operation frequencies.

| Dataset | if | while | for | and | or | except | return | break | continue | bool_op |
|---|---|---|---|---|---|---|---|---|---|---|
| Code Alpaca | 0.42 | 0.06 | 0.43 | 0.03 | 0.01 | 0.01 | 0.66 | 0.01 | 0.00 | 0.05 |
| Evol CodeAlpaca | 1.35 | 0.15 | 0.68 | 0.15 | 0.14 | 0.12 | 1.59 | 0.03 | 0.02 | 0.29 |
| CodeFeedBack | 1.59 | 0.14 | 0.76 | 0.19 | 0.14 | 0.08 | 1.62 | 0.05 | 0.02 | 0.33 |
| OSS Instruct | 1.38 | 0.07 | 0.59 | 0.16 | 0.08 | 0.07 | 1.58 | 0.04 | 0.01 | 0.24 |
| Ours (func-level) | 2.29 | 0.16 | 1.14 | 0.16 | 0.20 | 0.70 | 3.06 | 0.05 | 0.10 | 0.35 |
| Ours (file-level) | **3.60** | **0.38** | **1.77** | **0.25** | **0.21** | **1.11** | **5.24** | **0.08** | **0.10** | **0.45** |

Table 14: Detailed metrics of code strictness complexity

| Dataset | Type Hints | Parameter Validation | Value Verification | Exception Handling | Assertions | Doc Strings | Return Value Validation |
|---|---|---|---|---|---|---|---|
| Code Alpaca | 0.00 | 0.00 | 0.04 | 0.02 | 0.00 | 0.06 | 0.07 |
| Evol CodeAlpaca | 0.21 | 0.08 | 0.14 | 0.20 | **0.03** | 0.01 | 0.15 |
| CodeFeedBack | 0.42 | 0.09 | 0.16 | 0.10 | 0.01 | 0.05 | 0.14 |
| OSS Instruct | **0.94** | 0.09 | 0.12 | 0.10 | 0.02 | 0.08 | 0.15 |
| Ours (func-level) | 0.94 | 0.10 | 0.29 | 0.81 | 0.02 | **2.45** | 0.34 |
| Ours (file-level) | 0.43 | **0.28** | **0.40** | **1.76** | 0.01 | 1.74 | **0.80** |

comprehensive and rigorous consideration to code documentation. Additionally, we demonstrate clear advantages in *exception handling*, *return value validation*, and *type hints*, suggesting that our data is more standardized and stringent.

These analyses upon Halstead complexity, strictness and cyclomatic complexity, collectively demonstrate that feature tree-based code synthesis can create code that is both more complex and more sophisticated than current synthesis methods.

C.3.2. PROMPTS FOR EVALUATING CODE COMPLEXITY USING GPT-4O.

We adopt the LLM-as-a-judge methodology, using GPT-4o to comprehensively evaluate code complexity across four key dimensions: Error Handling, Modularity, Data Structure Complexity, and Third-Party Dependencies. We define four distinct levels of standards for each dimension and strategically leverage GPT-4o to assign a precise score to each sample based on these well-defined criteria. Detailed evaluation criteria, corresponding prompts, and scoring methodology are shown below.

---

**Prompts for Grading Data Structure Usage Complexity.**

You are an expert in evaluating complexity levels of data structure implementations for given code. Please provide a single integer score from 2 to 8.

**Criteria:**
Score 2 for case:
Basic data types only; Only use primitive types (int, string, etc.); Simple arrays or lists; No custom data structures.

Score 4 for case:
Basic data structures; Uses built-in collections (sets, maps); Simple combinations of basic structures; Basic object-oriented classes.

Score 6 for case:
Intermediate data structures; Custom data structures for specific needs; Efficient combination of multiple structures; Clear interfaces for data access.

Score 8 for case:
Advanced data structures; Specialized tree/graph structures; Optimized for operation requirements; Well-designed structure relationships

**Inputs:**
You are judging the following code:
## Begin Code ##
{code}
## End Code ##

**Output Format:**
Please provide your evaluation in the following format:
Grade: a single integer within 2, 4, 6, and 8

---

**Prompts for Grading Modularity Implementation Complexity.**

You are an expert in evaluating code architecture complexity, with a special focus on modularity implementation patterns. Please provide a single integer score from 2 to 8.

**Criteria:**
Score 2 for case:
Code without any modular designs, all logic in a single file with no clear separation.

Score 4 for case:
Code with minimal modularization, basic separation of logic but with tight coupling.

Score 6 for case:
Code with reasonable modularization and logical separation (e.g., some decoupling and partial adherence to design patterns, but limited scalability or reuse).

Score 8 for case:
Code implementation with clear and practical modularization (e.g., modules have distinct responsibilities, dependencies are simple and direct, and logical layers such as database, business logic, and user interface are separated. The design is easy to read, maintain, and extend for most real-world needs).

**Inputs:**
You are judging the following code:
## Begin Code ##
{code}
## End Code ##

**Output Format:**
Please provide your evaluation in the following format:
Grade: a single integer within 2, 4, 6, and 8

**Prompts for Grading Third Party Dependency Complexity.**

You are an expert in evaluating third-party dependency complexity of given code. Please evaluate the complexity of third-party library usage and dependencies in the following code and provide a single integer score from 2 to 8.

**Criteria:**
Score 2 for case: No external library usage, only built-in modules (e.g., os, sys, json).
Score 4 for case: Uses single third-party library with basic function calls (e.g., pandas, scipy, numpy).
Score 6 for case: Uses 2-3 third-party libraries.
Score 8 for case: Uses at least three third-party libraries with some interaction between them.

**Inputs:**
You are judging the following code:
## Begin Code ##
{code}
## End Code ##

**Output Format:**
Please provide your evaluation in the following format:
Grade: a single integer within 2, 4, 6, and 8.

**Prompts for Grading Error Handling Complexity.**

You are an expert in evaluating error handling complexity of given code. Please provide a single integer score from 2 to 8.

**Criteria:**
Score 2 for case: Complete lack of error handling.
Score 4 for case: Basic error handling that prevents crashes.
Score 6 for case: Basic error handling that prevents crashes with informative logging info.
Score 8 for case: Error handling covers major scenarios.

**Inputs:**
You are judging the following code:
## Begin Code ##
{code}
## End Code ##

**Output Format:** Please provide your evaluation in the following format:
Grade: a single integer within 2, 4, 6, and 8.

Table 15: Distribution of total features across 1k samples.

| Datasets | Workflow | Implementation Style | Functionality | Resource Usage | Computation Operation | Security | User Interaction | Data Processing |
|---|---|---|---|---|---|---|---|---|
| Alpaca | 1842 | 926 | 1005 | 324 | 525 | 8 | 181 | 331 |
| CodeFeedback | 3718 | 1018 | 1260 | 560 | **1432** | 50 | 379 | 1354 |
| Evol-Alpaca | 3550 | 1013 | 1305 | 598 | 1290 | 60 | 325 | 1838 |
| OSS-Instruct | 3106 | 1015 | 1192 | 421 | 585 | 49 | 278 | 1163 |
| Ours (func-level) | 4004 | 1050 | 1671 | 831 | 1213 | **227** | 542 | **3436** |
| Ours (file-level) | **4663** | **1244** | **2629** | **1183** | 778 | 123 | **1407** | 3160 |

| Datasets | File Operation | Error Handling | Logging | Dependency Relations | Algorithm | Data Structures | Implementation Logic | Advanced Techniques | Avg. |
|---|---|---|---|---|---|---|---|---|---|
| Alpaca | 18 | 77 | 1 | 166 | 365 | 1309 | 1143 | 17 | 8.24 |
| CodeFeedback | 46 | 421 | 13 | 453 | **719** | 1624 | 1770 | 81 | 14.90 |
| Evol-Alpaca | 85 | 354 | 15 | 811 | 579 | 1661 | 1480 | 135 | 15.10 |
| OSS-Instruct | 236 | 394 | 73 | 799 | 170 | 1804 | 1453 | 30 | 12.77 |
| Ours (func-level) | 567 | 887 | 205 | 2499 | 451 | 2297 | 1866 | 143 | 21.89 |
| Ours (file-level) | **1453** | **1029** | **548** | **4010** | 454 | **2565** | **2187** | **212** | **27.65** |

## C.4. Diversity Evaluation

### C.4.1. PROMPT FOR FEATURE EXTRACTION.

**Prompts for feature extraction.**

Extract features from the provided code snippets, following the requirements for each category below, formatted in JSON structure.
Responses in the following categories should be concise and organized in a JSON format surrounded with <begin> and <end>. Categories may include nested structures if applicable. Here is an example of the expected format:
<begin>{
"functionality": [ "data processing" ],
"computation operation": { "mathematical operation":[ "find common divisor", "base conversion", "prime factorization" ], "statistical calculations":[ "maximum" ] },
"data processing": { "data transformation": [ "drop rows" ] },
"data structures": [ "string", "list", "graph", "tree" ],
"implementation logic":["conditional", "loop"]
}<end>

**Categories to extract:**
1. **Programming Language:** Note the programming language used. Example: ["Python", "Java"].
2. **Workflow:** Outline the main steps and operations the code performs. Example: ["data loading", "preprocessing", "model training", "evaluation", "results saving"].
3. **Implementation Style:** What programming paradigm the code follows. Example: ["procedural", "object-oriented", "functional"].
4. **Functionality:** Explain the function of the code. Example: ["data processing", "user interaction", "system control"].
... ...
16. **Advanced Techniques:** Specify any sophisticated algorithms or methods applied. Example: ["Machine Learning", "Linear Regression", "Optimization Algorithms"].

**Requirements:**
1. If the code snippet contains fewer than three lines, only extract the most precise and relevant single feature.
2. For a function, provide no more than five features, prioritizing the most critical and distinctive aspects.
3. Only evaluate the code snippet; disregard any natural language descriptions or comments outside the code context.
4. Try not to let a feature appear in multiple categories at the same time.

**Inputs:**
{source_code}
**Output Format:**
<begin>
"workflow": ["your answer"],
"implementation style": ["your answer"],
"functionality": ["your answer"],
"resource usage": ["your answer"],
"computation operation": ["your answer"],
"security": ["your answer"],
"user interaction": ["your answer"],
"data processing": ["your answer"],
"file operation": ["your answer"],
"error handling": ["your answer"],
"logging": ["your answer"],
"dependency relations": ["your answer"],
"algorithm": ["your answer"],
"data structures": ["your answer"],
"advanced techniques": ["your answer"],
<end>
If the features of a category cannot be directly extracted from the code, please set it to an empty list [].

### C.4.2. TOTAL FEATURE DIVERSITY COMPARISON.

We compared the number of features in each category, as shown in Table 15. On average, our function-level data contains 21.89 features per sample, while the file-level data includes 27.65 features per sample. This demonstrates that our dataset not only leads in unique features but also significantly surpasses others in total feature count.

