# OpenReview forum: "EpiCoder: Encompassing Diversity and Complexity in Code Generation"
_ICML.cc/2025/Conference — ICML 2025 poster_

### Official Review · Reviewer_Sgyp · 2025-02-25

**Overall Recommendation:** 3

**Summary:**

The paper introduces *EpiCoder*, a novel approach to enhancing code-generation performance through hierarchical feature trees extracted from seed code. Empirical results demonstrate that EpiCoder surpasses similarly sized baselines in functional correctness (measured via pass@k) and complexity (measured using Halstead complexity and LLM-as-a-Judge). Additionally, the authors explore EpiCoder's potential to handle code generation tasks from function-level generation to multi-file and repository-level code synthesis.

**Claims And Evidence:**

Claims made in this submission are supported by experimental evidence.

**Essential References Not Discussed:**

The key contribution of this paper is a feature tree-based code generation algorithm. It only cites common code LLMs and code synthesis datasets. There are several more relevant works [1, 2] should be discussed.

[1] Li, H., Zhou, X., & Shen, Z. (2024). Rewriting the code: A simple method for large language model augmented code search. arXiv preprint arXiv:2401.04514.

[2] Koziolek, H., Grüner, S., Hark, R., Ashiwal, V., Linsbauer, S., & Eskandani, N. (2024, April). LLM-based and retrieval-augmented control code generation. In Proceedings of the 1st International Workshop on Large Language Models for Code (pp. 22-29).

**Experimental Designs Or Analyses:**

Correct to me.

**Methods And Evaluation Criteria:**

Correct to me.

**Other Comments Or Suggestions:**

- **Definition of 'Feature' and 'Feature Tree':** Since this is a feature tree-based code generation framework, providing a clear, upfront definition of “feature” and “feature tree” would greatly improve the paper’s clarity and readability.

* **Overview illustration (Figure 2)**:
    * **A) Feature Tree Extraction**: Could you provide what the raw code samples look like in your code set? Also, please clarify what the 'blue box' represents and how 'clustered features' differ from 'extracted features'.
    * **B) Feature Tree Evolution**: Since the notion of “feature” remains somewhat unclear, please specify if any restrictions or guidelines govern how new feature nodes are added during evolution.
    * **C) Code Generation**: If the two histograms represent feature distribution or frequency, it would be helpful to show the exact steps from the feature tree (Section B) to the final code (Section C). Additionally, please explain Docker’s role—is it an execution sandbox? If so, you should include some configuration details. The folder tree in the code block is also somewhat confusing; more explanation would be beneficial.
    * **Overall**: The overview could be made clearer by indicating what each stage takes as input and produces as output, along with more explicit navigation through the framework.

**Other Strengths And Weaknesses:**

## Strengths
- **Feature Tree–Based Code Synthesis**:
The hierarchical “feature tree” approach captures fine-grained code features (e.g., data structures, control flows), enabling adjustable levels of complexity in code generation.

- **Promising Performance**:
The authors synthesize 433k instruction data and train EpiCoder, which achieves state-of-the-art performance among comparably sized models in multiple function-level and file-level benchmarks.

- **Extensive Experimental Validation**:
The authors conduct extensive experiments to compare their proposed method with baselines, demonstrating their approach outperforms others in terms of Complexity and Diversity.

## Weaknesses
- **Complex and Resource-Intensive Pipeline**:
Constructing and evolving a feature tree, then repeatedly refining the generated code through test-and-debug cycles, can be intricate and computationally expensive. An alternative approach—retrieving and filtering real-world code—may offer greater efficiency and control to provide high-quality training data.

- **Potential for Hallucinations or Overfitting**:
Although feature tree–based synthesis can increase the diversity and complexity of generated code, the reliance on evolved features may introduce distribution shifts from the 'real-world' code. This can lead to hallucinations or overfitting, potentially compromising real-world code generation quality.

**Questions For Authors:**

- My main concern is about the purpose of the feature tree. In the code generation pipeline, you sample features from the subtrees into a set and feed it to an LLM. This raises the question of whether the tree structure is necessary. Could you clarify why you need such a complex pipeline to construct a tree rather than a simple frequency-based representation?

**Relation To Broader Scientific Literature:**

This paper builds on existing research in **LLM-driven code generation** by introducing a feature tree–based framework that systematically captures hierarchical semantic relationships. It is Inspired by **abstract syntax trees (ASTs)** yet extends beyond snippet-based approaches. By clustering features and enabling control over depth and breadth, their method moves beyond simple function-level tasks to complex multi-file or repository-level code.

**Theoretical Claims:**

No theoretical claims in this paper.

---

> ### Author Rebuttal · Authors · 2025-04-01
>
> Thank you for your thorough review and valuable feedback on our paper.
>
> ## 1. Additional References
> We appreciate your suggestion and will include discussions on them in our paper. Li et al. (2024) explore the use of LLMs for rewriting code to enhance code search performance, while Koziolek et al. (2024) propose a retrieval-augmented method for controlled code generation.
>
> ## 2. Necessity of Synthetic Data
> Certain types of data are extremely scarce in real-world scenarios, making synthetic data an essential and widely adopted approach in both academia and industry. For example, Qwen2.5-Coder [1] utilizes tens of millions of synthetic instruction samples, and models like DeepSeek-V3 [2] and R1 [3] also incorporate synthetic data during training. For code instruction data, while collecting raw code is relatively easy, obtaining well-structured tasks along with their corresponding solution code is significantly more challenging. Other works also adopt synthetic code instruction data, such as WaveCoder [4], MagiCoder [5], SelfCodeAlign [6], etc.
>
> While potential of hallucinations are inherent challenges in any synthetic data approach, we have implemented several measures to mitigate these issues, including frequency-based distribution adjustment, verification through test cases, and enhancing complexity and diversity. Therefore, we believe these concerns are not specific weaknesses of our method, but rather common challenges faced by all synthetic data approaches. In addition, recently there is an interesting work by Stanford University [6] on studying which is the critical factor of model’s benefitting from synthesized data. The main conclusion shows that the presence of reasoning behaviors, rather than correctness of answers, proves to be the critical factor, which suggests that although hallucinations do exist in the synthesized data, these data may still improve the models because of some other factors we must not neglect.
>
> Regarding the domain shifting risk of feature tree evolution, we must admit that every coin has two sides.
>
> For the positive side, we curate feature trees from seed data, which has been under certain preprocessing pipeline and will be unable to capture the entire domain distribution of “real word data”. Thanks to the evolution of feature trees, such innate problems can be alleviated.
>
> For the negative side, evolution on feature tree will involve noises.
>
> However, to alleviate the problem of evolution noises, we apply filtering in feature combination when generating new questions. Furthermore, the results of training on our synthesized data can present consistent improvements across multiple benchmarks, which proves that the benefits outweigh the drawbacks.
>
> ## 3. Definition of Feature and Feature Tree
> Features refers to abstractions of code. We organize features into a tree structure based on their logical relationships, where parent and child nodes represent a hierarchical containment relationship. To aid understanding, we provide illustrative examples in Figure 1(a) and Appendix C.
>
> ## 4. Clarifications on Figure 2
> ### Example of Raw Code Sample
> You can refer to the dataset at bigcode/the-stack-v2 on Hugging Face. https://huggingface.co/datasets/bigcode/the-stack-v2
>
> ### Difference Between Clustered Features and Extracted Features
> The key difference lies in how they are generated. Extracted features are directly obtained during the extraction process, while clustered features are introduced during clustering to ensure structural completeness. For example, in Figure 1(a), we extract features like "computation," "XOR," and "AND." During clustering, we introduce the clustered feature "Logical Operation" to connect them in a meaningful hierarchy.
>
> ### Evolution Process
> We use an LLM to guide this process. Relevant prompts and detailed examples can be found in Appendix C.3.
>
> ### Code Generation
> An example of the full generation process is provided in Appendix C.4 and C.5.
>
> ### Docker
> The Docker is an execution sandbox. We will open-source our code along with relevant configuration details.
>
> ## 5. Advantages of the Tree Structure
> We first clarify a misunderstanding: our method does not sample individual features from subtrees but instead samples entire subtrees, ensuring compatibility during code generation. And please refer to our response to reviewer eNxJ to see the clarification on the advantages of the tree structure.
>
> ## References
> [1] "Qwen2.5-Coder Technical Report." arXiv 2409.
>
> [2] "DeepSeek-V3 Technical Report." arXiv 2412.
>
> [3] "DeepSeek-R1: Incentivizing Reasoning Capability in LLMs via Reinforcement Learning." arXiv 2501.
>
> [4] Yu Z, et al. "WaveCoder: Widespread And Versatile Enhancement For Code Large Language Models By Instruction Tuning." ACL2024.
>
> [5] Wei Y, et al. "Magicoder: Empowering Code Generation with OSS-Instruct." ICML2024.
>
> [6] Kanishk Gandhi, et al. "Cognitive Behaviors that Enable Self-Improving Reasoners, or, Four Habits of Highly Effective STaRs" arXiv 2503.

---

### Official Review · Reviewer_eNxJ · 2025-03-12

**Overall Recommendation:** 3

**Summary:**

This paper presents EpiCoder,  a novel framework designed for code generation, addressing the limitations of existing methods that rely on code snippets as seed data. It introduces a feature tree-based synthesis approach that captures hierarchical code features, enhancing complexity and diversity in generated code. By refining a structured feature tree, EpiCoder allows precise control over code synthesis, supporting function-level and multi-file scenarios. Extensive experiments demonstrate that EpiCoder-trained models achieve state-of-the-art performance on multiple benchmarks.

**Claims And Evidence:**

Yes.

**Essential References Not Discussed:**

No.

**Experimental Designs Or Analyses:**

Yes.  Epi-coders are evaluated on several common code benchmarks.

**Methods And Evaluation Criteria:**

Yes.

**Other Comments Or Suggestions:**

No.

**Other Strengths And Weaknesses:**

**Strengths**
1. The authors propose a novel method to synthesize code instruction fine-tuning data.
2. This method could generate diverse Instruction data, furthermore, it could be adopted to repo-level code generation

**Weaknesses**
1. The method of constructing a feature tree is very complicated, and the description in the main text is not detailed enough and difficult to understand.
2. Many key parameters for building the tree are missing.  It’s difficult to reproduce or follow without code.

**Questions For Authors:**

1. What do you think are the advantages of building features through trees compared to directly extracting independent features?
2. When generating task data, how to ensure the rationality of the provided feature combination, for example, there are some contradictory features.

**Relation To Broader Scientific Literature:**

This work is related to code generation for large language models and fine-tuning code instructions for llms.

**Theoretical Claims:**

N/A

---

> ### Author Rebuttal · Authors · 2025-04-01
>
> Thank you for your valuable feedback. We address your concerns below and hope these clarifications help resolve them.
>
> ## 1. Missing Supplementary Material
> Our paper includes a 26-page appendix at the end, which provides extensive details on our methodology, implementation, and experiments.
>
> ## 2. Implementation Details
> Appendix C includes a step-by-step breakdown with detailed prompts and execution examples for better understanding. Additionally, we plan to **open-source our code and data** as soon as possible (in a month) while adhering to anonymity policies, ensuring full reproducibility. We hope this addresses your concern.
>
> ## 3. Rationality of Feature Combination
> As detailed in Appendix C.4 (line 1663-1664), our approach ensures that the LLM selects mutually compatible feature subsets when generating tasks. This guarantees that all generated data samples maintain a reasonable and coherent combination of features.
>
> ## 4. Advantages of the Feature Tree
> Our feature tree is constructed based on real code, leveraging structured modeling of the hierarchical relationships between code features to explicitly capture semantic associations among code elements. Furthermore, by utilizing the hierarchical topology of the feature tree, we can synthesize new data that has not appeared in real-world scenarios (seed code corpus), thereby expanding the model's generalization boundary for complex code patterns.
>
> As mentioned in the introduction (line 70-84), the key advantages are:
>
> ### (1) Controllable Complexity
> We control complexity by adjusting the tree's shape, such as depth and width. In contrast, using independent features relies solely on increasing feature count, which often leads to incompatible features or unnatural combinations that do not reflect real-world scenarios.
> Section 3.4 (Figure 5) and Appendix A.3 show the effectiveness of feature tree for generating complex file-level and repo-level code data.
>
> ### (2) Targeted Learning
> For example, if we need to generate some data focused on data processing, we can adjust the distribution to increase the probability of sampling the node "data processing" and its subnodes. This structured relationship is not possible with independent features, where it is unclear which features are related to "data processing".
>
> ### (3) Evolution Efficiency
> The tree structure provides clear and organized directions (depth and breadth) for evolution, making the process more efficient and achieving broader coverage.  An example is shown in Appendix C.3.
>
> Besides, in our response to Reviewer AwKV, we have added the comparison with SelfCodeAlign [1], which follows an approach closer to independent feature-based methods. The result is shown in Table 2 of [`https://anonymous.4open.science/r/epicoder_rebuttal-C619/tables.md`](https://anonymous.4open.science/r/epicoder_rebuttal-C619/tables.md), and the 4% improvement highlights the effectiveness of our approach.
>
> ---
>
> ## References
> [1] Wei Y, Cassano F, Liu J, et al. "SelfCodeAlign: Self-Alignment for Code Generation." *NeurIPS*, 2024.

---

### Official Review · Reviewer_n7np · 2025-03-14

**Overall Recommendation:** 3

**Summary:**

This paper introduces EpiCoder, a feature tree-based framework for code generation that addresses diversity and complexity in generated code. The authors propose a hierarchical feature to represent features like concepts used in the code. The framework consists of three components: (1) Feature Tree Extraction, where features are extracted from seed data and organized into a tree structure, (2) Feature Tree Evolution, which iteratively expands the tree to increase diversity beyond the seed data, and (3) Feature Tree-Based Code Generation, which samples from the tree to create code conditioned on a sampled subtree. The generated code can range from single function to file levels. They not only show good performance on function-level benchmarks including HumanEval, MBPP, and BigCodeBench, but they also created a new file-level benchmark XFileDep to evaluate the performance of their method, showing great performance compared to other open models.

**Claims And Evidence:**

Their claim that hierarchical feature trees can enable the generation of more complex and diverse code is supported by the experimental results. The authors demonstrate performance improvements on multiple standard benchmarks and provide analysis of complexity metrics compared to existing approaches. They also show that their approach can handle multiple levels of complexity, from function-level to file-level generation with the newly proposed XFileDep benchmark.

**Essential References Not Discussed:**

Most of the related work is discussed.

**Experimental Designs Or Analyses:**

The experiments for the function-level code follow the standard coding benchmarks. The experiments for the file-level code are run on their own proposed XFileDep benchmark.

**Methods And Evaluation Criteria:**

They address the train/test leakage problem by using the EvoEval benchmark to show that their method is not overfitting the benchmark data. Although the score on EvoEval showing the model has a similar score to closed models like Claude 3 is a bit strange.
The proposed XFileDep benchmark seems like a good benchmark for file-level code generation. There are details about the benchmark design in the appendix. One issue is that since it uses a similar synthetic generation pipeline, it may unfairly favor their own model as the finetune data may be more aligned with the distribution of this benchmark dataset.

**Other Comments Or Suggestions:**

Please see the above sections for some comments and suggestions.

**Other Strengths And Weaknesses:**

The paper is well written and the method is novel and simple. The experimental results are comprehensive. The main weakness is that the function-level benchmark may include a train/test leakage problem, and I don't think the EvoEval as a synthetic benchmark is a gold standard to fully address this issue. The file-level benchmark is very interesting, but it being synthetic data coming from a very similar pipeline may give unfair advantage to the proposed method.

**Questions For Authors:**

N/A

**Relation To Broader Scientific Literature:**

The work is related to many code synthetic data generation methods and finetune work like WaveCoder and Evol-Instruct, etc.

**Theoretical Claims:**

N/A: The paper is an empirical studies on synthetic data generation for code and finetuning LLMs.

---

> ### Author Rebuttal · Authors · 2025-04-01
>
> Thank you for your thoughtful feedback and for recognizing the novelty and comprehensiveness of our work. We address your concerns below.
>
> ## 1. Train/Test Leakage in Function-Level Benchmarks
> We address train/test leakage analysis in Appendix B.2 (Figure 9), demonstrates that EpiCoder has a low risk of data leakage. We further validates our low potential leakage in EvoEval and you can view the updated figure at the following anonymous link:
> [`https://anonymous.4open.science/r/epicoder_rebuttal-C619/leakage_analysis.png`](https://anonymous.4open.science/r/epicoder_rebuttal-C619/leakage_analysis.png)
>
> ## 2. Potential Bias in the XFileDep Benchmark
> The details of the XFileDep construction process are in Appendix A.2. To mitigate potential bias, we have implemented the following measures:
>
> - **Pipeline Difference**: More filters and human checks make the distribution differ.
> - **Data Isolation**: The benchmark data is strictly separated from the training data to prevent direct overlap.
> - **Similarity Filtering**: We apply similarity-based filtering to remove benchmark data that exceeds the leakage threshold according to embedding similarity, reducing potential bias.
> - **Data Format Difference**:
>   - In Supervised Fine-Tuning (SFT), our model generates the entire code given the task.
>   - In the benchmark, the model generates code based on the key class/function name/docstring in the file.
>
> We believe these measures help ensure that XFileDep serves as a fair and valuable benchmark for evaluating file-level code generation.

---

### Official Review · Reviewer_AwKV · 2025-03-20

**Overall Recommendation:** 3

**Summary:**

This paper presents a new data synthesis method to generate complex and diverse code data. Given some seed code data, this method prompts an LLM to extract code features (e.g., functionality concepts, programming paradigm, etc.) from each code and organize them into a tree structure (i.e., feature tree). It then prompts the LLM to expand the tree by introducing new concepts in the same category or sub-categories. The authors used this method to generate 380K code functions and 53K code files. Using this new data, they finetuned base LLMs to obtain EpiCoder series and demonstrated that these models achieved better performance than existing models on five benchmarks.

## update after rebuttal
I want to thank the authors for conducting the additional experiments based on my suggestion. My major concern about the unfair comparison has been addressed by the new comparison results using datasets with comparable sizes. The new comparison between EpiCoder and SelfCodeAlign is also helpful.

On the other hand, while the authors claimed that the newly proposed benchmark is a more challenging/better benchmark than ClassEval etc., it would be more convincing if the authors could evaluate EpiCoder on some existing benchmarks since they are peer-reviewed and widely used. I also suggest the authors increase the number of manually analyzed data points in the test case analysis. 30 data points seem not enough.

Nevertheless, my major concerns have been addressed. So I am happy to raise my score from 2 to 3.

**Claims And Evidence:**

The claims about the technical approach and novelty are largely reasonable. To the best of my knowledge, SelfCodeAlign is the only work that shares a similar idea of extracting high-level code features (or code concepts as called by the authors of SelfCodeAlign) from seed data and generating new data based on the extracted features. Compared with SelfCodeAlign, this work considers more kinds of code features, represents the extracted features in a nice and clean tree structure, and proposes a new component to enrich the features in the tree structure. So I think this work is novel enough.

- Wei, Yuxiang, et al. "SelfCodeAlign: Self-Alignment for Code Generation." The Thirty-eighth Annual Conference on Neural Information Processing Systems.

My main concern is about the claims on the effectiveness of the data generated by the proposed method. The authors compared EpiCoder with existing models like MagiCoder and WaveCoder and attributed the better performance of EpiCoder to the higher complexity and diversity of their synthetic data. However, this is a fair comparison, since the EpiCoder data is much bigger than the synthetic data used by MagiCoder and WaveCoder. Specifically, the EpiCoder data includes 380K code functions and 53K code files, while the MagiCoder data only includes 75K code functions, and the WaveCoder data only includes 111K code functions. It is likely that the better performance of EpiCoder is simply because of the significantly larger finetuning dataset. Furthermore, the performance improvement over the baseline models is not significant (2-3% compared to the second-best baseline). This makes it less convincing that the feature-based synthesis method is really effective.

**Essential References Not Discussed:**

As discussed in my comments to claims, a very related method is SelfCodeAlign. The authors should discuss this work and compare EpiCoder with SelfCodeAlign.

- Wei, Yuxiang, et al. "SelfCodeAlign: Self-Alignment for Code Generation." The Thirty-eighth Annual Conference on Neural Information Processing Systems.

**Experimental Designs Or Analyses:**

As mentioned above, a major issue of the evaluation is the unfair comparison to the baseline models like MagiCoder and WaveCoder. The authors need to make their synthetic dataset the same size as the baselines to eliminate this confounding factor. Otherwise, it is not convincing that EpiCoder's better performance is really due to the diversity and complexity of the synthetic dataset.

Since SelfCodeAlign also leverages code concepts to generate data and achieves good performance, it is a more related and state-of-the-art method to compare with.

The authors constructed a new benchmark called XFileDep for file-level code generation. Since there are many known class-level or repo-level code generation benchmarks, it is questionable why the authors chose to create a new benchmark instead of using existing benchmarks, especially given that XFileDep is not carefully evaluated. The authors should evaluate EpiCoder on at least one known and peer-reviewed class-level or repo-level benchmark.

In Section 4.1.2, the authors used GPT-4o to estimate code complexity in four dimensions. There is no evaluation of the accuracies of GPT-4o. If GPT-4o is not very accurate, it doesn't make much sense to analyze or interpret the results generated by GPT-4o.

**Methods And Evaluation Criteria:**

The proposed method makes sense. I do have some concerns about LLM hallucinations since this method makes heavy use of LLMs for feature extraction, clustering, and evolution. Yet other methods like OSS-Instruct also makes use of LLMs and it also seems LLMs are able to learn meaningful patterns from noisy data. So I don't think this is a major issue. But some analysis or manual validation would be helpful.

A more significant issue is about the feature sampling and code generation steps in the proposed method. Currently, this method performs weighted sampling over the features in the tree and then prompts the LLM to generate code based on the sampled features. It does not consider whether it makes sense to put some features together when generating code. So it may generate code with potentially irrelevant or even contradicting features. The generated code may be syntactically correct but doesn't make much sense in practice. This may be a potential reason why the synthetic dataset does not gain a significant performance improvement even though its size is 3-4 times bigger than other datasets. I suggest the authors sample some generated code functions/files and conduct a manual analysis to check whether they make sense.

The last step of the data synthesis method is about iterative refinement. However, it is questionable whether the test files generated by the proposed methods are indeed effective. It is likely that the LLM generates some weak or even invalid test files. While these tests are executable, they do not really examine the generated code properly.

**Other Comments Or Suggestions:**

The order of the appendencies is confusing. It would be easier to follow if the authors first showed the prompts for each component in the data synthesis method, followed by additional experiments and results.

**Other Strengths And Weaknesses:**

None

**Questions For Authors:**

1. How do you know that the better performance of EpiCoder is because of data complexity and diversity instead of data size?

2. How many generated code functions/files have a reasonable combination of features?

3. Why not use existing class-level or repo-level code generation benchmarks?

4. How many generated test cases are valid?

5. What is the accuracy of GPT-4o in estimating code complexity?

**Relation To Broader Scientific Literature:**

The proposed method can be inspirational to data synthesis research beyond code generation.

**Theoretical Claims:**

There are no theoretical claims in this paper.

---

> ### Author Rebuttal · Authors · 2025-04-01
>
> Thank you for your detailed feedback and for recognizing the contribution of our work. Below, we address your key concerns.
> The table link [https://anonymous.4open.science/r/epicoder_rebuttal-C619/tables.md](https://anonymous.4open.science/r/epicoder_rebuttal-C619/tables.md) contains all the tables referenced in our responses.
> ## Questions
> ### Q1: How do you know that the better performance of EpiCoder is due to data complexity and diversity instead of dataset size?
> A: We acknowledge that data complexity, data diversity, and dataset size all contribute to performance improvements. To isolate the effect of dataset size, we randomly sampling 75K data from EpiCoder and comparing the results with MagiCoder (75K) and WaveCoder (20K + 110K). The results, presented in Table 1 of the table link, show 5.4% and 3.0% improvements respectively.
>
> ### Q2: How many generated code functions/files have a reasonable combination of features?
> A: Our data generation process incorporates constraints to ensure feature compatibility, as detailed in Appendix C.4 (line 1663-1664).  When generating tasks, the LLM selects a subset of features that are mutually compatible.
>
> ### Q3: Why not use existing class-level or repo-level code generation benchmarks?
>
> A: Existing benchmarks present several limitations:
>
> - Class-Level Benchmarks (e.g., ClassEval)
>   - These are essentially function-level generation or completion tasks, a simplification of our file-level code generation.
>
> - Repo-Level Benchmarks face two major issues:
>   - Metrics such as Exact Match (EM) and Edit Similarity (ES) are only suitable for base models and not for content-rich instruct models (e.g., CrossCodeEval, RepoBench).
>   - EvoCodeBench's codebase showing focuses mainly on the local context (in the same file) and does not take into account the dependencies of the whole repository.
>
> Our benchmark  fully encompasses complete tasks and generation, covering various instruction formats, making further elaboration unnecessary. Instead of focusing on function-level tasks, we prefer to evaluate the LLM’s ability to generate complete files based on natural language instructions.
>
> ### Q4: How many generated test cases are valid?
>
> - Manual Evaluation:  We manually examined 30 data samples and found that all the generated test cases correctly reflected the task requirements. However, we observed that these test cases tend to be relatively simple and may not cover all edge cases. A concrete example is provided in Appendix C5.2 and C5.3.
>
> - Pass Rate Improvement: According to our statistics, before the first refinement iteration, only 32% of the generated code passed all test cases. After three iterations, the pass rate increased to 61%, demonstrating that the test cases effectively filter out low-quality data and improve overall data quality, even if they do not guarantee 100% correctness.
>
> ### Q5: What is the accuracy of GPT-4o in estimating code complexity?
>
> A: To estimate the accuracy of GPT-4o in assessing code complexity, we conducted pairwise comparisons on a dataset using both human evaluation and GPT-4o, showing an average win rate of 84.4% and 74% for our data and a consistency of 79.375% between human evaluation and GPT-4o.
>
> In section 4.1, we use both GPT-4o and the sofware engineering method to estimate code complexity, and both approaches consistently indicate the better complexity of our data. We also assessed complexity using DeepSeek-V3-0324 and Llama3.1-70B-Instruct, following the same metrics as in Section 4.1.2. The results align with our original conclusions. Detailed results are in Table 3-6 of the table link.
>
> ## Other Comments
>
> ### Comparison with SelfCodeAlign
> Thank you for acknowledging our contribution. We will discuss SelfCodeAlign in our paper. As you mentioned, the key difference between EpiCoder and SelfCodeAlign is that EpiCoder organizes features into a tree structure and uses evolution to enrich features.  To make a fair comparison, we first sample a subset of our data with same size of SelfCodeAlign and then finetune CodeQwen-Base for comparing with the results in their paper. Table 2 of the table link shows that our data has a 4% improvement.
>
> ### Performance Improvement
> As you mentioned, we have achieved a 2-3% improvement compared to the second-best baseline, which is already a notable gain at the instruction tuning stage. Additionally, the second-best baseline Qwen2.5-Coder-7B-Instruct utilizes tens of millions of synthetic instruction samples as stated in section 4.2 of their technical report, which demonstrate the effectiveness of our data.
>
> ### Manual Check
> During the whole pipeline, we incorporated manual inspection and optimization to ensure that the generated code is reasonable and coherent. We provide concrete examples in Appendix C.
>
> ### Order of Appendices
> Thanks for your valuable suggestions and we will adjust it accordingly.
>
> We hope our responses effectively resolve your concerns.

---

### Decision · Program_Chairs · 2025-05-01

**Decision:**

Accept (poster)

**Comment:**

This paper presents EpiCoder, which uses a novel feature-tree based synthesis framework to generate more diverse and complex synthetic  data as well as allows for targeted learning by adjusting probabilities of features. The fine-tuned models on these synthetic datasets achieve better performance on many benchmarks. All reviewers appreciated the overall simple and novel idea of using feature tree to represent and generate synthetic code examples for better diversity and complexity, and the experiments show promising improvements. However, there were also some concerns around the complexity of the approach, unfair comparisons with baselines, potential of contradictory features during generation, XFileDep being a bit more biased benchmark, hallucinations during data generation and potential data leakage. The author response helped alleviate many of these concerns. It would be great to incorporate the discussions and detailed feedback from reviews in the final version of the paper.